# A Path-Planning Method Considering Environmental Disturbance Based on VPF-RRT*

Zhihao Chen [1], Jiabin Yu [1,2,*], Zhiyao Zhao [1,2], Xiaoyi Wang [3] and Yang Chen [1]

[1] School of Artificial Intelligence, Beijing Technology and Business University, Beijing 100048, China
[2] Beijing Laboratory for Intelligent Environmental Protection, Beijing Technology and Business University, Beijing 100048, China
[3] School of Arts and Sciences, Beijing Institute of Fashion Technology, Beijing 100029, China
* Correspondence: yujiabin@th.btbu.edu.cn

**Abstract:** In the traditional rapidly exploring random tree (RRT) algorithm, the planned path is not smooth, the distance is long, and the fault tolerance rate of the planned path is low. Disturbances in an environment can cause unmanned surface vessels (USVs) to deviate from their planned path during navigation. Therefore, this paper proposed a path-planning method considering environmental disturbance based on virtual potential field RRT* (VPF-RRT*). First, on the basis of the RRT* algorithm, a VPF-RRT* algorithm is proposed for planning the planning path. Second, an anti-environmental disturbance method based on a deep recurrent neural networks PI (DRNN-PI) controller is proposed to allow the USV to eliminate environmental disturbance and maintain its track along the planning path. Comparative simulation experiments between the proposed algorithm and the other algorithms were conducted within two different experimental scenes. In the path-planning simulation experiment, the VPF-RRT* algorithm had a shorter planning path and a smaller total turning angle when compared to the RRT* algorithm. In the path-tracking simulation experiment, when using the proposed algorithm, the USV could effectively compensate for the impact of environmental disturbance and maintain its navigation along the planning path. In order to avoid the contingency of the experiment and verify the effectiveness and generality of the proposed algorithm, three experiments were conducted. The simulation results verify the effectiveness of the proposed algorithm.

**Keywords:** unmanned surface vessel; path planning; rapidly exploring random tree algorithm; path tracking; diagonal recurrent neural networks; PI controller





## 1. Introduction

In recent years, unmanned surface vehicles (USVs) have been widely used for the water quality sampling of public water [1–4]. Disturbances exist in the real water environments, which is not conducive to the navigation safety of the USV [5–8]. Therefore, scholars have carried out a lot of research on USV path planning and path tracking methods under a disturbed water environment [9–12]. The hybrid problem of path planning and path tracking is the research focus of USV navigation technology [13–16].

At present, there are various path-planning methods. Path planning methods mainly include the A* algorithm [17], Dijkstra algorithm [18–20], rapidly exploring random trees (RRT)- [21], and deep reinforcement learning (DRL)-based algorithms [22]. The planning path of most traditional path-planning algorithms contains many unnecessary turning points, which is not applicable to the movement of USVs. In a real lake environment, there are environmental disturbances, such as wind and water flow, which lead to that the USV deviating from the planning path during navigation.

In order to solve the problem of path planning in environments with obstacles, a path-planning method considering environmental disturbances based on the virtual potential field RRT* is proposed. The main contributions of this work can be summarized as follows:

(1) A VPF-RRT* algorithm is proposed that has a short planning-path distance and high-fault tolerance rate.

(2) An anti-environmental disturbance method based on a DRNN-PI controller is proposed. This method can overcome the impact of environmental interferences on a USV. At the same time, the combination of DRNN and the PI controller improves the flexibility of the PI controller, which can adjust the parameters $K_p$ and $K_i$ in real-time.

This paper is organized as follows. Section 2 presents the related works. Section 3 provides the modeling and problem formulation of this paper. Section 4 presents the algorithm proposed in this paper. The experimental results are discussed in Section 5, and the paper is concluded finally in Section 6. This paper has strong novelty in the field of USV cruise. This paper optimizes the RRT* algorithm to improve the efficiency of path planning. At the same time, this paper improves the PI controller and uses the DRNN algorithm to adjust the parameters of the PI controller.

## 2. Related Works

### 2.1. Path Planning

In order to solve this problem, Zhang [23] proposed a new heuristic function combined with the artificial potential field (APF) method and introduced this into A* algorithm, which effectively reduces the turning points in the planning path and makes the path smoother. Yu [24] improved the expansion mode of the eight connections of the traditional A* algorithm to the 20 connections so that the sharpness of turning (at a corner) can be greatly reduced and the planning path is smoother. However, the calculation speed of the above algorithm is too slow. In the path-planning algorithm, the RRT algorithm has the advantage of a fast sampling speed, so it is widely used to solve the multitarget path-planning problem. However, the path length generated by the RRT algorithm is long. In order to solve this problem, Verbari [25] proposed a decentralized iterative algorithm based on single-agent dynamic RRT-star; this algorithm introduces a decentralized strategy that is based on an iterative plan-compare-assign process to reduce the length of the planning path. Park [26] proposed a boundary RRT* algorithm by increasing real-time performance through simple calculations and using the boundary of the configuration space. The functions of the above algorithms are shown in Table 1.

**Table 1.** The algorithm and simulation experimental parameters.

| Algorithm | Path Smoothing | Increase of Efficiency |
|---|---|---|
| Autonomous land vehicle path-planning algorithm based on improved heuristic function of A-Star [23] | Yes | / |
| Improved safety-first A-star algorithm for autonomous vehicle [24] | Yes | / |
| Multiagent trajectory planning: A decentralized iterative algorithm based on single-agent dynamic RRT star [25] | / | Yes |
| Boundary-RRT* algorithm for drone collision avoidance and interleaved path replanning [26] | / | Yes |

### 2.2. Path Tracking

In order to solve this problem, Li [27] proposed a new fractional-order PID controller with the desired gain and phase margin, which increases the accuracy and robustness of the system and enhances the control accuracy and antidisturbance ability of the USV. Wu [28] proposed an antidisturbance nonlinear control law that can accurately track the target in a limited time based on the red interference observer, which enhances the antidisturbance ability of the USV. Xu [29] proposed an improved double-factor adaptive Kalman filter to adjust the position and pose of a USV to achieve the purpose of antidisturbance. Zhang [30] proposed an antidisturbance control law for a USV to make the yaw angle and position of the USV reach the expected value and reduce the impact of environmental disturbance. Peng [31] proposed a data-driven adaptive antidisturbance control method to estimate the unknown input gain, unmeasured speed, and total disturbance and combined this with a

PID controller to reduce the impact of environmental disturbance. The above algorithms are only based on the yaw and deviation of the USV, which does not consider the environmental disturbance as an input of the controller to control the USV. The parameters of a traditional linear controller need to be manually set and cannot be adjusted in real-time, lacking flexibility. In order to solve this problem, Yao [32] proposed a PID controller based on a fuzzy algorithm and a BP neural network, which can adjust three parameters in the PID controller in real time, improve the flexibility of the PID controller, and reduce the yaw of a USV during navigation. Xu [33] proposed a backstepping-based controller design for uncertain, switched high-order nonlinear systems via PI compensation, which combines a backstepping algorithm with a PI controller and introduces a feedback neural network to adjust the parameters in real-time; thus, the flexibility of PI controller is improved. The neural network selected by the above algorithms is too simple to effectively deal with dynamic problems. The functions of the above algorithms are shown in Table 2.

**Table 2.** The algorithm and simulation experimental parameters.

| Algorithm | Improve Anti-Interference Capability | Increase Flexibility |
|---|---|---|
| Fractional-order controller for course-keeping of underactuated surface vessels based on frequency domain specification and improved particle swarm optimization algorithm [27] | Yes | / |
| Antidisturbance leader–follower synchronization control of marine vessels for underway replenishment based on robust exact differentiators [28] | Yes | / |
| An antivibration-shock inertial matching measurement method for hull deformation [29] | Yes | / |
| Antidisturbance control for dynamic positioning system of ships with disturbances [30] | Yes | / |
| Output-feedback flocking control of multiple autonomous surface vehicles based on data-driven adaptive extended state observers [31] | Yes | / |
| Research and comparison of automatic control algorithm for unmanned ship [32] | / | Yes |
| Backstepping-based controller design for uncertain Switched high-order nonlinear systems via pi compensation [33] | / | Yes |

Based on the above defects, this paper proposed a path-planning method considering environmental disturbances based on a virtual potential field RRT* (VPF-RRT*). First of all, a VPF-RRT* algorithm is proposed for path planning. The virtual potential field is introduced into the RRT* algorithm to adjust the position of the path node, connecting the starting point and the target point. The connecting line segment generates the attractive potential field, which is superimposed with the repulsive potential field generated by the obstacle. The position of the path node is adjusted according to the size of the superimposed potential field. Second, aiming at the problem of the USV devices being moved from the planning path caused by environmental disturbance, an anti-environmental disturbance method based on a deep recurrent neural networks PI (DRNN-PI) controller is proposed. The yaw angle was obtained by detecting the environmental disturbance through the sensor. The DRNN-PI controller was used to obtain the compensation angle and correct the yaw angle so that the USV could overcome the environmental disturbance and navigate on the planning path. At the same time, the DRNN neural network was combined with a PI controller, and a DRNN neural network was used to select the two parameters $K_p$ and $K_i$, which allowed the PI controller to adjust the parameters in real time. The implementation process of the algorithm is shown in Figure 1.

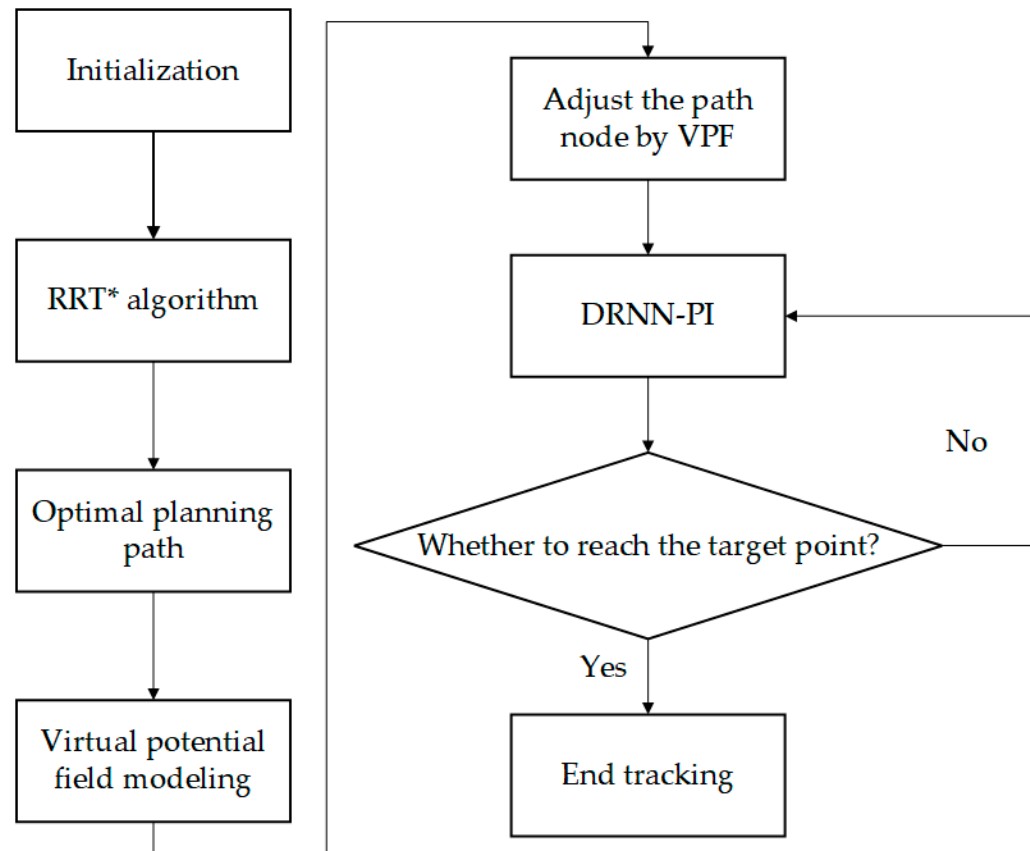

**Figure 1.** The path planning method considering environmental disturbance based on VPF-RRT* flowchart.

### 3. Modeling and Problem Formulation

*3.1. USV Modeling*

This study defines the USV movement co-ordinate system on the Earth's surface, as shown in Figure 2. A fixed co-ordinate system is established on the Earth's surface, where the origin $O_e$ of the Earth co-ordinate system can be any point on the Earth's surface, and the $Y_e$- and $X_e$-axes point to the north and east of the Earth, respectively, thus, this system can be simplified as an {e} co-ordinate system. In order to simplify the analysis, this paper considers only the surge, sway, and yaw of a USV. The three-degrees-of-freedom mathematical model of a USV is expressed as follows:

$$\begin{cases} \dot{\eta} = J(\eta)v \\ M \cdot \dot{v} + C \cdot v + D \cdot v = \tau + \omega \end{cases} \tag{1}$$

where $\eta = [x, y, \psi]^T$ is the pose vector, which defines the position and heading angle; $v = [u, v, r]^T$ is the speed vector, which includes the forward, sway, and steering speeds; $J(\eta)$ is the transformation matrix from the Earth co-ordinate system to the USV motion co-ordinate system. The second equation denotes a dynamic equation, where $\tau$ is the torque vector, $\omega$ is the force vector defined by the wind, wave, and current in an environment, $M$ is the inertia matrix of a USV, $C$ is the Coriolis centripetal matrix, and $D$ is the resistance matrix [34].

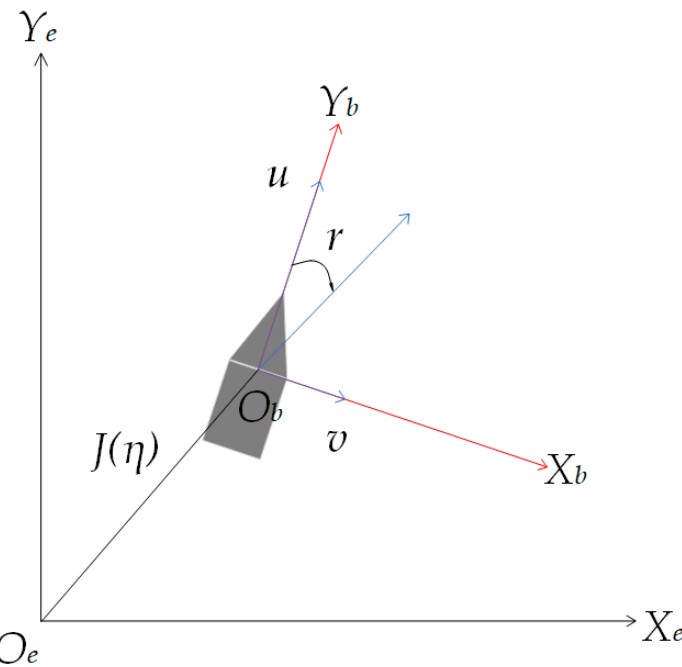

**Figure 2.** The USV movement co-ordinate system.

A real USV was modeled by the Unity3D platform to obtain a 3D USV model. The real USV and its 3D model are presented in Figure 3a,b, respectively. The collision and collision detection modules of the Unity3D platform were added to the 3D USV model to facilitate the observation of the collision of the 3D USV model. This paper adds rigid body and kinematic rigid body collider to the 3D USV model to ensure the inertia and gravity of the model. Finally, the buoyancy module is added to the 3D USV model. The abovementioned modeling methods can make the 3D USV model have real physical properties in simulation experiments, and the proposed algorithm can be more effectively verified in the simulation experiment.

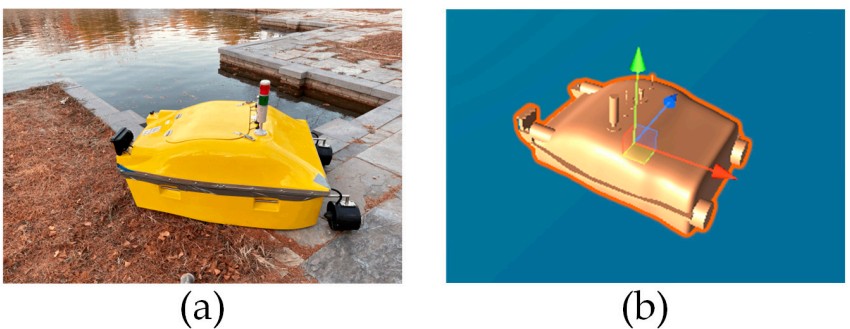

(a)          (b)

**Figure 3.** The USV: (**a**) the real USV; (**b**) the 3D USV model.

### 3.2. Obstacle Modeling

This paper uses the Unity3D platform to model the obstacles in the map. Since a USV is navigated only on the horizontal plane, the section of an obstacle on the horizontal plane is regarded as an obstacle in the path-planning process. Due to the uncertainty of obstacles on the water surface and to ensure the navigation safety of a USV, it is necessary to leave a safe distance between a USV and the obstacles and conduct an expansion modeling of the obstacles. The obstacle co-ordinate system {o} is established with the center of the obstacle as the origin, as shown in Figure 4. In Figure 4a, the static obstacle before expansion can be seen, and in Figure 4b is the static obstacle after expansion.

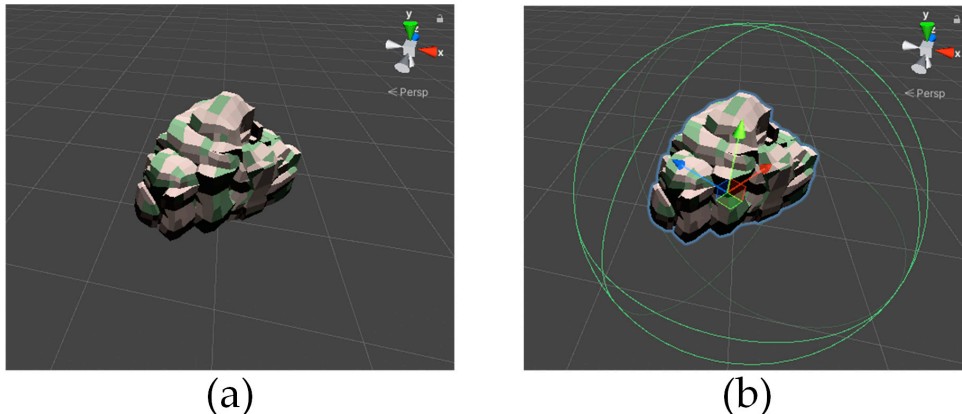

**Figure 4.** The obstacle: (**a**) static obstacle; (**b**) static obstacles after expansion.

The boundary co-ordinates of the obstacles are $(x_{o.i}, y_{o.i})$, $i = 1, 2, \ldots, n$. The boundary of the obstacle is expanded, and the expression is as follows:

$$\left(\frac{x^*_{o.i} - x_o}{a + B}\right)^q + \left(\frac{y^*_{o.i} - y_o}{b + B}\right)^p = 1 \tag{2}$$

where $B$ is the obstacle expansion coefficient, and $x^*_{o.i}$ and $y^*_{o.i}$ are the expanded obstacle boundary co-ordinates. $q$ and $p$ are the gains of the equivalent geometry of the obstacles on the $x$- and $y$-axes in the $\{e\}$ co-ordinate system, respectively. $x_o$ is the central co-ordinates of the obstacle on the $x$-axis, and the expression is as follows:

$$x_o = \frac{\sum\limits_{i=1}^{n} x_{o.i}}{n} \tag{3}$$

and $y_o$ is the central co-ordinates of the obstacle on the $y$-axis, and the expression is as follows:

$$y_o = \frac{\sum\limits_{i=1}^{n} y_{o.i}}{n} \tag{4}$$

### 3.3. Problem Formulation

This paper requires the USV to navigate to the end-point in the shortest distance on the basis of ensuring safety. Therefore, it is necessary to establish an evaluation standard based on distance. The planning path $P$ is established by connecting a series of path nodes or grids. The Euclidean distance of the planning path $P$ is:

$$D(P) = \sum_{i=1}^{k-1} s_i, s_i = \sqrt{(x_{i+1} - x_i)^2 + (y_{i+1} - y_i)^2} \tag{5}$$

where $s_i$ is the Euclidean distance between adjacent path nodes or grids, and $(x_i, y_i)$, $i = \{1, 2, \ldots, k - 1\}$ is the co-ordinate of each path node or grid.

There are various obstacles in the lake environment, such as reefs and islands. These obstacles constitute the forbidden area $O$ that USV must avoid for voyage safety. First, on the premise that the path can avoid the forbidden area $O$, the shortest path can be obtained by comparing the length $D(P)$ of different paths. Based on the above, build an objective function:

$$\underset{P}{\arg\min}[D(P)], P \cap O = \varnothing \tag{6}$$

where $D(P)$ is the Euclidean distance of the planning path $P$, and the planning path $P$ does not pass through the forbidden area $O$.

## 4. Materials and Methods

### 4.1. A Virtual Potential Field -RRT* Algorithm

The RRT algorithm is an incremental construction method. During the construction process, the algorithm continuously generates a random state point in the search space. If the point is in a collision-free position, the nearest node in the search tree is found as the reference node. From the reference node, it extends towards the random node with a certain step length. The location of the end-point of the extension line is added to the search tree as an effective node. The growth process of the search tree continues until the distance between the target node, and the search tree is within a certain range. Then the search algorithm searches for the shortest path connecting the starting point to the end-point in the search tree. The traditional RRT algorithm is a path-planning algorithm with a random sampling tree structure. The basic idea of an RRT algorithm is to construct paths by adding specific random trees. The algorithm does not need to model space, and the expansion between nodes does not need to be preprocessed. This algorithm will detect the collision of paths between nodes to solve the obstacle avoidance problem. This algorithm can quickly search the blank area in the space and further reduce the distance between the path node and the target point [35].

When the distance between the planning path and the obstacle is close, the fault tolerance rate of the planning path will be reduced, which is not conducive to the subsequent path tracking of USV. The smooth planning path is more suitable for the path tracking of USV, but there are many unnecessary inflection points in the planning path of the traditional RRT* algorithm, so it is necessary to adjust the nodes to optimize the angle of inflection points. However, the existing path smoothing algorithm will increase the memory loss of the algorithm. Therefore, it is necessary to adjust the path node and use the VPF-RRT* algorithm to plan the path of USV. The expansion step is p, the initial value of *i* is 1, and the maximum number of iterations is *n*.

Connect the start point and the target point. This line is the virtual gravity lines. Obstacles will generate repulsive virtual potential fields. Adjust the position of path nodes according to the combined virtual potential fields. This method shortens the long planning path, adjusts the distance between path nodes and obstacles, and improves the fault tolerance rate of the path, as shown in Figure 5.

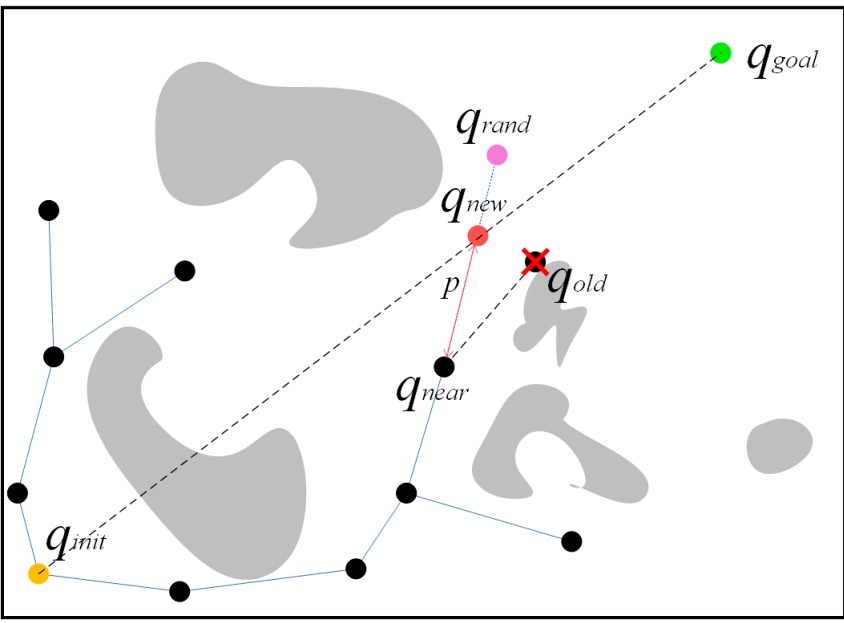

**Figure 5.** The expansion process of the random sampling points of the VPF-RRT* algorithm.

Make a vertical line from the path node to the gravity line, and the length of the vertical line is $L$. Calculate the gravity $F_y$ of the current gravity line on the node; its expression is as follows:

$$F_y = -F_z \cdot \sin \alpha \tag{7}$$

where $\alpha$ is the included angle between the planning path and the gravitational line; $F_z$ is the tension of the planning path. Let the maximum angular velocity of USV be $\omega$, the mass of USV is derived from $m$ and inertia coefficient $B$, and the expression is as follows:

$$F_z = \frac{m\omega^2 L}{B} \tag{8}$$

Let the repulsive force of the obstacle on the node be $F_c$, and its expression is as follows:

$$F_c = \begin{cases} \eta m\omega_{\max}^2 R_{\min}^3 \left(\frac{1}{L} - \frac{1}{\rho}\right)^2, & L < \rho \\ 0, & L \geq \rho \end{cases} \tag{9}$$

where $\eta$ is the repulsion coefficient, whose value is set manually; $\omega_{\max}$ is the maximum angular velocity of USV; $R_{\min}$ is the minimum turning radius of USV; $\rho$ is the range of influence of obstacles on nodes. When the path node exceeds the influence range of the obstacle, the repulsive force is zero.

Using the difference between the gravitational force $F_y$ and the repulsive force $F_c$ in the current situation, the gravitational difference $F_y^*$ is obtained, and its expression is as follows:

$$F_y^* = F_y - F_c \tag{10}$$

Use $F_y^*$ to calculate the compensation distance $\Delta L$. The compensation distance $\Delta L$ will be used to adjust the position of the path node, and its expression is as follows:

$$\Delta L = \frac{-E F_y^*}{m\omega^2 \sin \alpha} \tag{11}$$

Use the difference between the current vertical segment $L$ and the compensation distance $\Delta L$ to obtain a new vertical segment $L$, whose expression is as follows:

$$L' = L - \Delta L \tag{12}$$

The pseudocode of VPF-RRT* is shown in Algorithm 1.

| **Algorithm 1**: VPF-RRT* | |
|---|---|
| Step1: | T ← InitializeTree (T, qinit); |
| Step2: | **For** i = 1: n do |
| Step3: | qrand ← Sample (T, qgoal, M); |
| Step4: | qnear ← Nearest (T, qrand); |
| Step5: | **End** qnew ← Steer (qnear, qrand, p); |
| Step6: | qneighbor ← Findnearneighbor (T, qnew, M); |
| Step7: | if CollisionFree (qnew, T, M) then |
| Step8: | T ← Chooseparent (qnew, qneighbor, T); |
| Step9: | T ← Rewire (T, qnew, qneighbor); |
| Step10: | T ← (12); |
| Step11: | **Return** T; |

### 4.2. An Anti-Environmental Disturbance Method Based on DRNN-PI Controller

The previous section introduced the VPF-RRT* algorithm, which is used to plan the path from the starting point to the target point. This section proposes an anti-environmental disturbance method based on a DRNN-PI controller, which is used to control USV navigation on the planned path and realize path tracking.

The recurrent neural networks can be divided into single hidden layer recurrent neural networks and deep recurrent neural networks (DRNN) according to the structure [36]. DRNN can enhance the ability of the PI controller, which can repeat the loop body at each time many times [37]. The parameters in the loop body of each layer of the DRNN network are shared, but the parameters between different layers can be different [38]. The DRNN network has strong sensitivity. DRNN stores the output value of the hidden layer neuron at the previous time through the receiving layer and returns it to the input of the first hidden layer so that the final output of the network is related to the current information and the historical information [39]. DRNN has strong dynamic information processing ability, high prediction accuracy, and strong generalization ability, which can avoid local minimum problems [40].

Set the input vector of the DRNN neural network as $x = [x_1, x_2, \ldots, x_m]$; there are n hidden layers, and the number of nodes in each layer is represented by $l_1, l_2, \ldots, l_k$, and the number of nodes in the output layer is 1. When the time is $t$, the output vector of the first hidden layer of the DRNN neural network is:

$$R_1(t) = f\{W_1[X(t), z(t)] + B_1\} \tag{13}$$

where $R_1(t)$ is the output of the first hidden layer. $W_1$ is the weight matrix and $B_1$ is the threshold matrix, both of which are between the input layer and the first hidden layer. Function $f$ is the transfer function of the hidden layer of DRNN. $z(t)$ is the feedback state vector [41]. The output of other hidden layers except for the first hidden layer of DRNN is:

$$R_L(t) = f\{W_L R_{L-1}(t) + B_L\} \tag{14}$$

where $R_L(t)$ is the output vector of the $L$ hidden layer. $W_L$ is the weight matrix, and $B_L$ is the threshold matrix, both of which are between the $L$-1 hidden layer and the $L$ hidden layer [42]. The output $y(t)$ of the output layer of DRNN is:

$$y(t) = g[W_{n+1} R_n(t) + B_{n+1}] \tag{15}$$

where $W_{n+1}$ is the weight matrix and $B_{n+1}$ is the threshold matrix, both of which are between the $n$ + 1 hidden layer and the output layer. Function g is the transfer function of the output layer of DRNN. The structure of DRNN is shown in Figure 6. The DRNN network includes one input layer, two hidden layers, and one output layer.

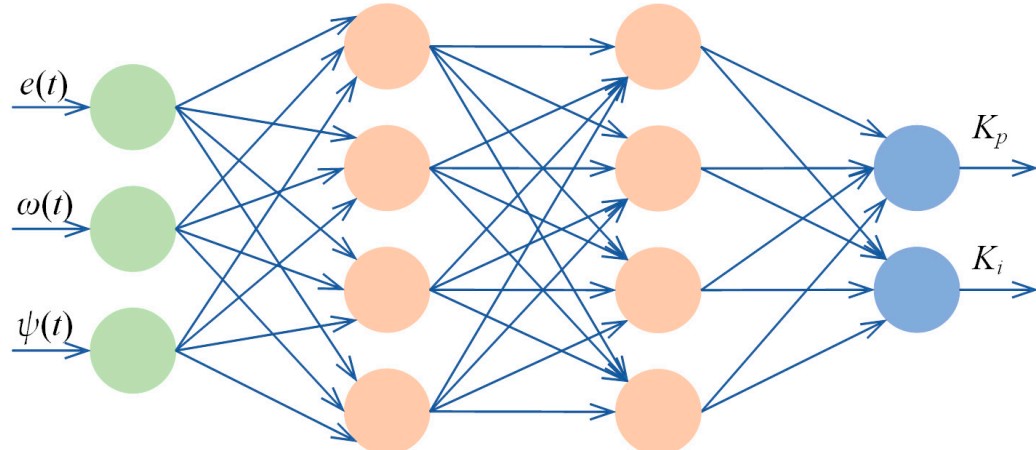

**Figure 6.** Structure of DRNN.

In the real lake environment, there are wind disturbances and wave disturbances. The USV deviates from the planning path due to the influence of wind and wave, which prolong

the actual sailing distance of the USV and increase the energy loss and the possibility of collision. Therefore, this paper proposes an anti-environmental disturbance method based on the DRNN-PI controller. First, the environmental disturbance information of the USV, such as wind and water flow, is obtained through the sensor. Second, a yaw angle is obtained by using the information on environmental disturbance, and the compensation angle is obtained by the DRNN-PI controller to correct the yaw angle so that the USV can overcome the influence of environmental disturbance. Finally, the USV navigates on the planning path.

The deep recurrent neural network (DRNN) network is used to adjust the performance of the PI controller. The hidden layer of the neural network has *L* nodes. There are *N* samples in each USV heading angle data set. The output of a single hidden layer is as follows:

$$\bar{s} = \frac{\sum\limits_{i=1}^{L} s_i}{L} \tag{16}$$

where $s_i$ is the output of a single hidden layer node. The DRNN neural network aims to minimize the error loss between the expected heading angle and the actual heading angle, as shown in the following:

$$\sum_{j=1}^{N} \|\bar{s}_j - \psi(t)\| = 0 \tag{17}$$

The loss function is used to obtain the output error of the network model, and the gradient descent method is used to find the minimum value, update the weight, and finally make the model converge. Mean square error (MSE) is selected as the loss function to measure the deviation between the predicted value and the real value after each step of training in the neural network. During the operation, the output value of the loss function will become smaller and smaller and eventually approaches zero. The expression of MSE is as follows:

$$MSE[\psi_d(t), \psi(t)] = \frac{1}{N} \sum_{i=1}^{N} [\psi_d(t) - \psi(t)]^2 \tag{18}$$

where $\Psi_d(t)$ is the expected heading angle, $\Psi(t)$ is the actual heading angle, and *N* is the total number of samples. In addition, the optimizer selects the root mean square propagation (RMSProp) method. The RMSProp method is suitable for dealing with nonstationary targets. The root mean square error (RMSE) and mean relative error (MRE) are used to verify the prediction accuracy of the neural network model. The expression is as follows:

$$RMSE[\psi_d(t), \psi(t)] = \sqrt{\frac{1}{N} \sum_{i=1}^{N} [\psi_d(t) - \psi(t)]^2} \tag{19}$$

$$MRE[\psi_d(t), \psi(t)] = \frac{1}{N} \sum_{i=1}^{N} \left| \frac{\psi_d(t) - \psi(t)}{\psi(t)} \right| \times 100\% \tag{20}$$

where $\Psi_d(t)$ is the expected heading angle, $\Psi(t)$ is the actual heading angle, and *N* is the total number of samples.

The USV in this paper is equipped with sensors on both sides, which can detect the environmental disturbance force on the ship body, including wind disturbance force $F_{wind}$ and wave disturbance force $F_{wave}$. The force analysis of USV is shown in Figure 7, in which the directions of $F_{wind}$ and $F_{wave}$ are randomly selected.

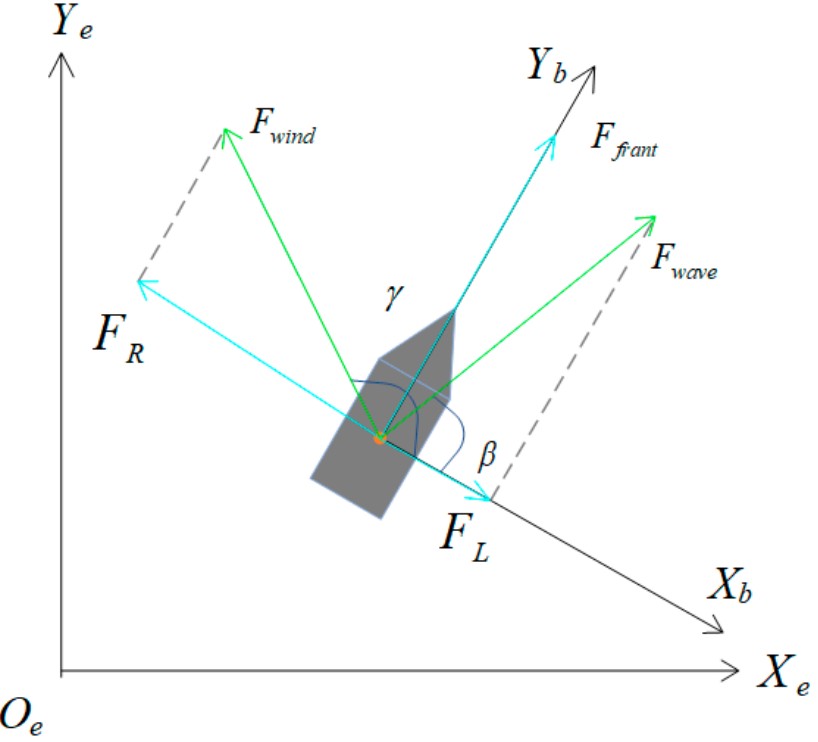

**Figure 7.** Force analysis of USV.

In Figure 7, the forward driving force $F_{frant}$ of the USV is constant. $F_R$ is the resultant force of environmental disturbance on the right side of the USV, and the direction of $F_R$ is consistent with the negative direction of $X_b$-axis. $F_L$ is the resultant force of environmental disturbance on the left side of the USV, and the direction of $F_L$ is consistent with the positive direction of $X_b$-axis. The expressions of $F_R$ and $F_L$ are:

$$\begin{cases} F_R = F_{wind} \cdot \cos(\pi - \gamma) + F_{wave} \cdot \cos(\pi - \beta) \\ F_L = F_{wind} \cdot \cos\gamma + F_{wave} \cdot \cos\beta \end{cases} \tag{21}$$

where $\gamma$ is the included angle between the wind disturbance force and the positive direction of the $X_b$-axis; $\beta$ is the included angle between the wave disturbance force and the positive direction of the $X_b$-axis. A yaw angle $\varepsilon(t)$ is obtained by using the environmental disturbance information. Then, set a compensation angle $\alpha(t)$, whose value is consistent with the yaw angle $\varepsilon(t)$. The formula is:

$$\alpha(t) = \varepsilon(t) = \left| \frac{\pi}{2} - \arccos\left(\frac{|F_R(t) - F_L(t)|}{F_f}\right) \right| \tag{22}$$

where, $F_R(t)$ is the resultant force of environmental disturbance on the right sides of the USV detected by the sensor at time $t$. $F_L(t)$ is the resultant force of environmental disturbance on the left sides of the USV detected by the sensor at time $t$. After the compensation angle $\alpha(t)$ is obtained, $\alpha(t)$ is introduced into the PI controller:

$$\omega(t) = K_p[\psi_d(t) - \psi(t)] + K_i \int [\psi_d(t) - \psi(t)]d\tau \tag{23}$$

where $\Psi_d(t)$ is the expected angle between the forward direction of the USV and the forward direction of the planning path. In order to resist the impact of environmental disturbance, $\Psi_d(t)$ needs to be equal to the compensation angle $\alpha(t)$, i.e., $\Psi_d(t) = \alpha(t)$. $\Psi(t)$ is the actual included angle between the forward direction of the USV and the forward direction of the planning path. $K_p$ is the proportional coefficient. $K_i$ is the integral coefficient. Let $F_{max}$ be

the maximum force that can yaw the USV; when $|F_R - F_L| > F_{max}$, the USV deviates from the planning path, as shown in Figure 8.

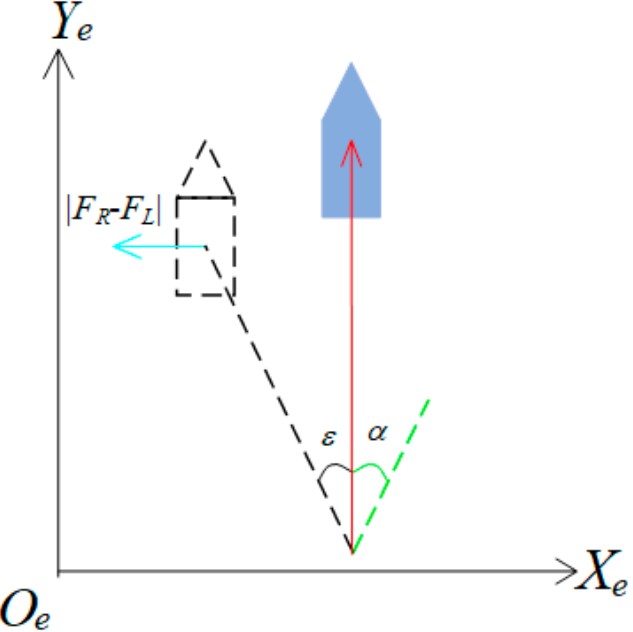

**Figure 8.** USV deviates from the planning path.

In Figure 8, the red line represents the planning path. $\varepsilon$ is the yaw angle obtained by using the environmental disturbance information, and $\alpha$ is the compensation angle. The black dotted line is the simulated path after the USV deviated from the planning path. When using the PI control to realize the control of the USV, the proportional coefficient $K_p$ and integral coefficient $K_i$ must be adjusted in real-time. There is a nonlinear relationship between $K_p$ and $K_i$. Traditional PI controllers adjust $K_p$ and $K_i$ manually, which leads to slow response speed and the poor anti-interference ability of PI controller. Therefore, a DRNN with strong self-learning abilities is combined with a PI controller to adjust $K_p$ and $K_i$ in this paper. In contrast to the traditional PI controller, the DRNN-PI controller adjusts $K_p$ and $K_i$ by obtaining the dynamic characteristics of the system in the internal feedback loop through the recursive nerve. The structure of DRNN-PI controller is shown in Figure 9.

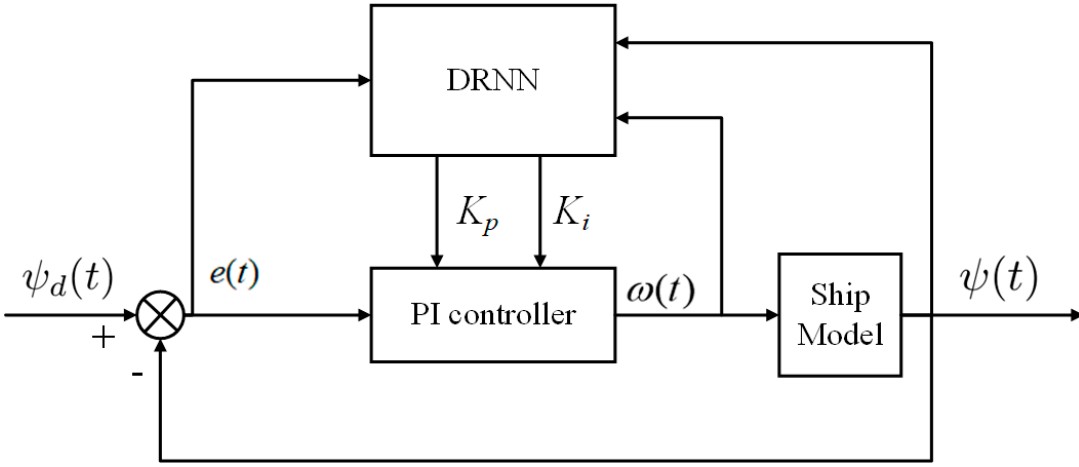

**Figure 9.** DRNN-PI controller structure.

In Figure 9, the expected included angle between the forward direction of the USV and the forward direction of the planning path is equal to the compensation angle, i.e., $\psi_d(t) = \alpha(t)$. The error $e(t)$ is the difference between the expected included angle $\psi_d(t)$ and the actual included angle $\psi(t)$, i.e., $e(t) = \alpha(t) - \psi(t)$. The integral separation is used to solve the integral saturation problem of the PI controller. When the integral is about to reach the saturation state, set the dead zone to limit the integral from reaching the saturation state. When integral saturation occurs, integral separation is used to eliminate integral saturation.

Before using the DRNN-PI controller, the parameters of DRNN need to be determined. First, the structure of the neural network is determined. In this paper, the number of input layer nodes is 3, the number of hidden layer nodes is 4, and the number of output layer nodes is 2. Second, each layer of the neural network is assigned an appropriate weight value and deviation value. In this paper, the initial weight value is 1, and the deviation value is 0. Finally, the activation function, learning rate, and inertia coefficient (suitable for the actual problem) are selected. In this paper, the ReLu function was used as the activation function of the neural network. The learning rate was 0.01, and the inertia coefficient was 0.1. The process of selecting proportional coefficient $K_p$ and integral coefficient $K_i$ by DRNN is shown in Algorithm 2:

| **Algorithm 2:** DRNN-PI controller | |
| :--- | :--- |
| Step1: | Initialize parameters; |
| Step2: | $\alpha(t) \leftarrow$ (22); |
| Step3: | $\psi_d(t) = \alpha(t)$; |
| Step4: | Sample $\psi_d(t)$ and $\psi(t)$; |
| Step5: | $\omega(t) \leftarrow$ (23); |
| Step6: | The inputs of DRNN $\leftarrow \psi_d(t)$, $\psi(t)$, $\omega(t)$; |
| Step7: | The outputs of DRNN $\leftarrow K_p$, $K_i$; |
| Step8: | DRNN starts iterative learning; |
| Step9: | **Return** $K_p$ and $K_i$; |

## 5. Simulation Experiment and Discussion

The simulation environment was based on the public water area of *Yuyuantan* Lake, located in the west of Beijing. The simulation experiment used Windows 10 as the operating system and Unity3D as the simulation tool. The hardware platform was an Intel Core i5-10200h processor with a main frequency of 2.4 GHz and 16 GB memory. In order to verify the effectiveness of the algorithm, a map of the simulation experiment was established based on Section 3. The parameters of the proposed algorithm and simulation experiment in this paper are presented in Table 3.

**Table 3.** The algorithm and simulation experimental parameters.

| Parameters | Definition | Numerical Value |
| :---: | :---: | :---: |
| $u$ (m/s) | Forward speed of USV | 10 |
| $F_{front}$ (N) | Forward force of USV | 30 |
| $r$ (rad/s) | Maximum angular velocity of USV | 0.3 |
| $R$ (m) | Minimum turning radius of USV | 3 |
| $a$ (m) | $x$-axis radius of obstacle | 8 |
| $b$ (m) | $y$-axis radius of obstacle | 8 |
| $q$ | $x$-axis gain of obstacle | 1.5 |
| $p$ | $y$-axis gain of obstacle | 1.5 |
| $R_s$ (m) | Safety range of USV | 6 |
| $F_{\max}$ (N) | Maximum disturbing force of USV | 10 |
| $B$ | Inertia coefficient | 8 |
| $\eta$ | Repulsion coefficient | 6 |

The parameter values in Table 1 are derived from experience. We created a 1000 m × 1000 m square 3D map containing some of the environmental disturbance areas. The co-ordinates of the center point of the square 3D map were (0,0). The upward direction along the center point on the map plane was the $Y_e$-axis, and the rightward direction along the center point on the map plane was the $X_e$-axis. The positions of obstacles on the map were random.

### 5.1. Simulation Experiment of Neural Network

We set a start point and a target point in an area containing obstacles and used the RRT algorithm to plan the path from the start point to the target point. The USV advanced at a speed of 10 m/s, and we used a DRNN-PI controller to control the steering of the USV. The DRNN neural network was trained based on the principle of the minimum difference between the actual heading angle and the expected heading angle of the USV. The simulation experiment was conducted in an obstacle-ridden environment. The BP and CMAC algorithms were used as comparison algorithms, and the error curves of the neural network are shown in Figure 10.

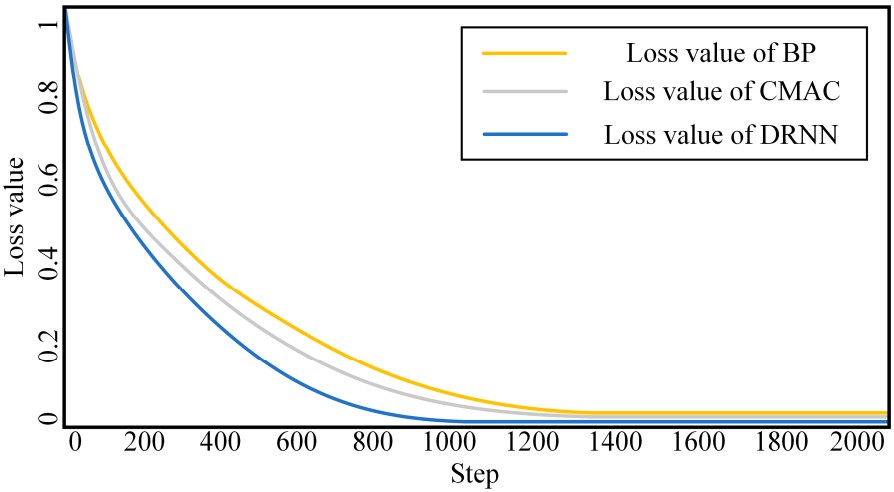

**Figure 10.** Simulation experiment results in obstacle environment.

It can be seen from Figure 10 that the convergence speed of the BP and CMAC algorithms is slower than that of the DRNN algorithm. The error curve of the BP algorithm tends to be stable only when the step in the iterations reaches about 1300. The error curve of the CMAC algorithm tends to be stable only when the step in the iterations reaches about 1200. When the DRNN algorithm iterates to about 1000 steps, the error curve starts to stabilize. Therefore, the convergence speed of the DRNN algorithm in the obstacle environment is higher than that of the comparison algorithm, and the loss value of the DRNN algorithm is lower.

The BP, CMAC, and DRNN network models were established. The network model parameters were set and trained 2000 times, respectively. The loss function value and computational time after running the program were recorded. A comparison of the results for the control accuracy of different models is shown in Table 4.

**Table 4.** Control accuracy of different models.

| Algorithm | RMSE | MRE | Loss | Computational Time (s) |
|---|---|---|---|---|
| BP-PI | 7.3519 | 12–14% | 0.0396 | 18.5181 |
| CMAC-PI | 6.5987 | 8–11% | 0.0287 | 16.2566 |
| DRNN-PI | 5.9182 | 7–9% | 0.0211 | 16.3271 |

Table 4 shows that, compared with BP and CMAC, the RMSE, MRE, and Loss of DRNN networks are lower. Because the hidden layer structure of DRNN is more complex, the computational time of DRNN is long, yet it is within a reasonable range, and the control accuracy of the DRNN is higher, so the effectiveness of the DRNN is better. The computational time refers to the difference between the sailing time obtained after the planning path length is divided by 10 m/s and the actual sailing time of the USV; that is, the time consumed by the PI controller to adjust the heading angle of the USV.

In order to avoid contingency in the experimental results, 20 groups of different starting points and targets were selected for the simulation experiment, and the results are shown in Figure 11.

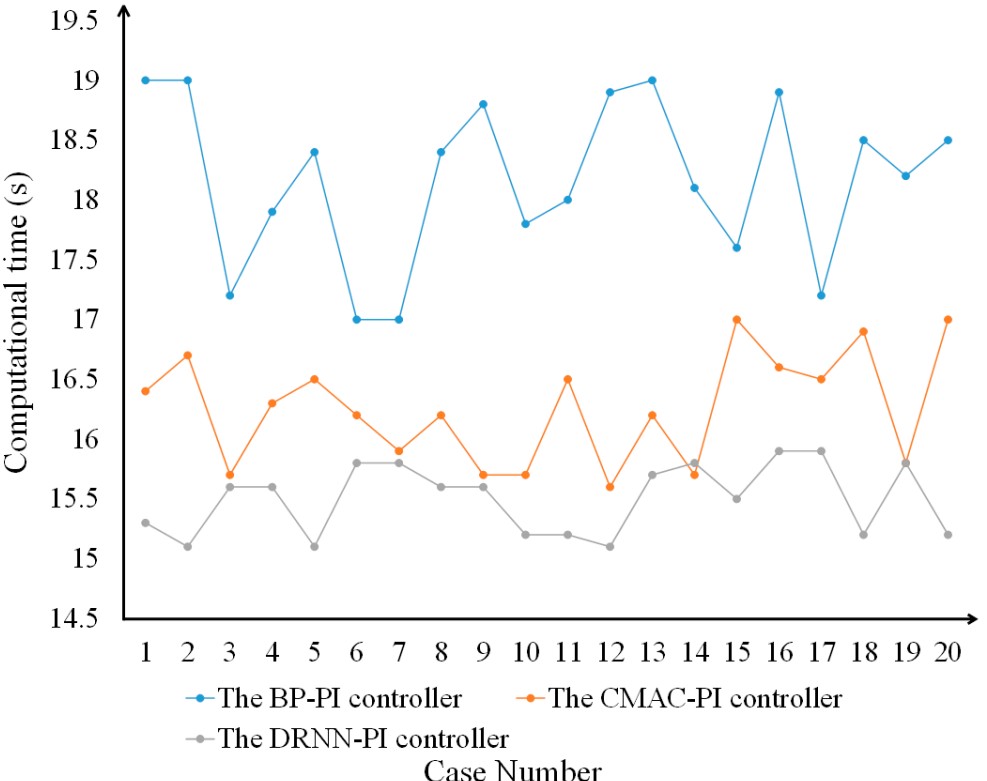

**Figure 11.** Experimental data of 20 groups for different starting points and targets.

It can be seen from Figure 11 that during the whole USV sailing process, the computational time required for the DRNN-PI algorithm (to adjust the USV heading angle) was between 15 s and 16 s, which was lower than the comparison algorithm. Although in the 14th and 19th experiments, the steps required by the proposed algorithm were similar to those of the comparison algorithm, the difference was within a reasonable range. Therefore, the simulation experiment proved that the ability to control the USV using the DRNN-PI algorithm was high and had strong practicability.

### 5.2. Simulation Experiment of Path Planning

In the path-planning simulation experiment, the co-ordinates of the start point were (423, 440), and the co-ordinates of the target point were (411, 411). Based on the above map parameters, the RRT* algorithm with a PI controller and the B-spline curve-RRT* with a PI controller was selected as the comparison. The comparison experiment was conducted with the RRT* algorithm with a PI controller, the B-spline curve-RRT* with a PI controller, and the VPF-RRT* algorithm with PI controller to verify the effectiveness of the planning path of the VPF-RRT* algorithm. The experimental results are shown in Figures 12–14 and Table 5.

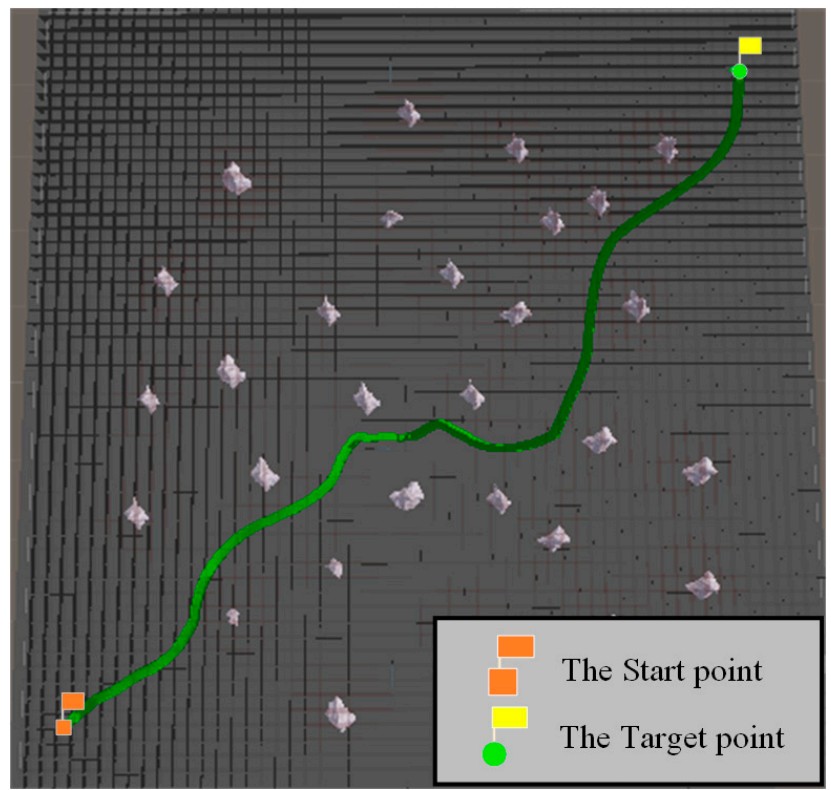

**Figure 12.** Results of RRT* algorithm with a PI controller in the path-planning simulation experiment.

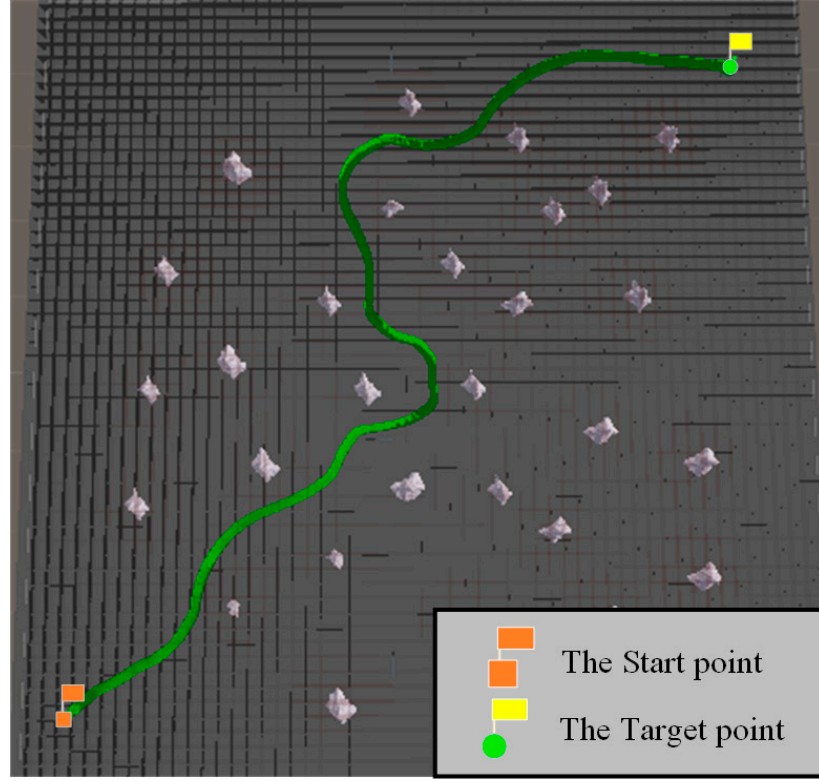

**Figure 13.** Results of B-spline curve-RRT* with a PI controller in the path-planning simulation experiment.

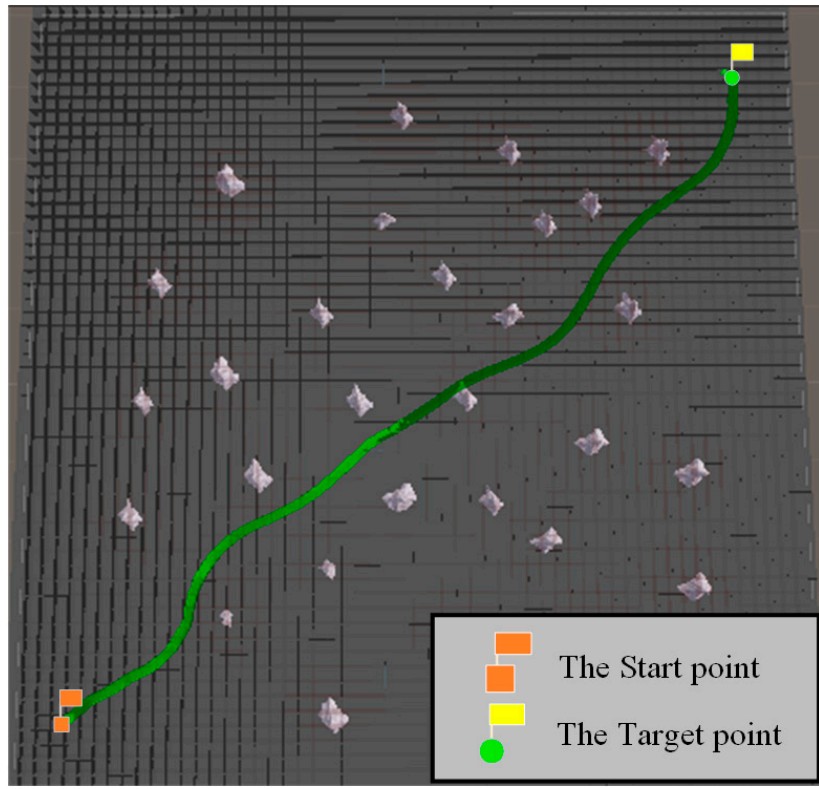

**Figure 14.** Results of VPF-RRT* algorithm with a PI controller in path planning simulation experiment.

**Table 5.** Results for path planning simulation experiment.

| Algorithm Name | Path Length (m) | Total Turning Angle | Computational Time (s) | Sailing Time (s) | Normalization Index |
|---|---|---|---|---|---|
| RRT* with PI controller | 1710.15182 | 161°3752′ | 22.15748 | 182.54843 | 1.43525 |
| B-spline curve-RRT* with PI controller | 1952.94236 | 286°1475′ | 26.22971 | 201.21855 | 1.63901 |
| VPF-RRT* with PI controller | 1623.07952 | 89°1398′ | 22.87613 | 178.33216 | 1.36217 |

When combined with the experimental results in Table 5, the normalized index value is the path length divided by the Euclidean distance between the start point and target point. It can be seen from Figure 12 that the length of the planning path of the RRT* algorithm with a PI controller is longer than the length of the VPF-RRT* algorithm with a PI controller, and the total turning angle of the RRT* algorithm with a PI controller is also larger than the total turning angle of the VPF-RRT* algorithm with a PI controller. It can be seen from Figure 13 that the length of the planning path of the B-spline curve-RRT* with a PI controller is longer than the length of the VPF-RRT* algorithm with a PI controller, and the total turning angle of the B-spline curve-RRT* with a PI controller is also larger than the total turning angle of the VPF-RRT* algorithm with a PI controller. As can be seen from Figure 14, since the VPF-RRT* algorithm adjusts the position of the path nodes based on the virtual potential field, this shortens the length of the planning path for the VPF-RRT* algorithm and reduces the total turning angle for it; the planning path of the VPF-RRT* algorithm was suitable for the movement ability of the USV.

In order to avoid the contingency of the experiment results, three groups of different start and target points were selected in the path-planning simulation experiments. The results are shown in Table 6.

**Table 6.** Results for the different start and target points under path planning simulation.

| Start and Target | Algorithm Name | Path Length (m) | Total Turning Angle | Computational Time (s) | Sailing Time (s) | Normalization Index |
|---|---|---|---|---|---|---|
| (−94, −27) (347, −451) | RRT* algorithm with PI controller | 725.15104 | 45°1268′ | 13.20158 | 90.12177 | 1.11500 |
| | B-spline curve-RRT* with PI controller | 816.20583 | 50°3288′ | 14.98402 | 99.37544 | 1.25501 |
| | VPF-RRT* algorithm with PI controller | 709.13587 | 44°1534′ | 13.81035 | 87.79353 | 1.09037 |
| (400, 400) (−400, −400) | RRT* algorithm with PI controller | 1634.98075 | 109°8418′ | 20.56508 | 171.34998 | 1.44513 |
| | B-spline curve-RRT* with PI controller | 2120.18598 | 302°9211′ | 26.30054 | 219.44503 | 1.87399 |
| | VPF-RRT* algorithm with PI controller | 1568.27831 | 105°7518′ | 18.60874 | 168.47056 | 1.38617 |
| (400, −400) (−400, 400) | RRT* algorithm with PI controller | 1612.83788 | 141°4894′ | 19.32007 | 182.21533 | 1.42556 |
| | B-spline curve-RRT* with PI controller | 1984.33276 | 291°0233′ | 24.11328 | 203.55983 | 1.75391 |
| | VPF-RRT* algorithm with PI controller | 1567.15489 | 134°4537′ | 18.83021 | 177.00357 | 1.38518 |

### 5.3. Simulation Experiment of Path Tracking

Environmental disturbance was added to the path-tracking simulation experiment. The parameters of disturbance are shown in Table 7. In Figure 15, the blue arrow is the wave disturbance, and the direction of the wave is consistent with the direction of the blue arrow. The white arrow is the wind disturbance, and the direction of the wind is consistent with the direction of the $Y_e$-axis.

**Table 7.** Environmental disturbance parameters.

| Parameters | Definition | Numerical Value |
|---|---|---|
| $F_{wind}$ (N) | Disturbing force of wind | 1 |
| $F_{wave}$ (N) | Disturbing force of wave | 15 |
| $v_{wind}$ (m/s) | Velocity vector of wind | 0.1 |
| $v_{wave}$ (m/s) | Velocity vector of wave | 3 |

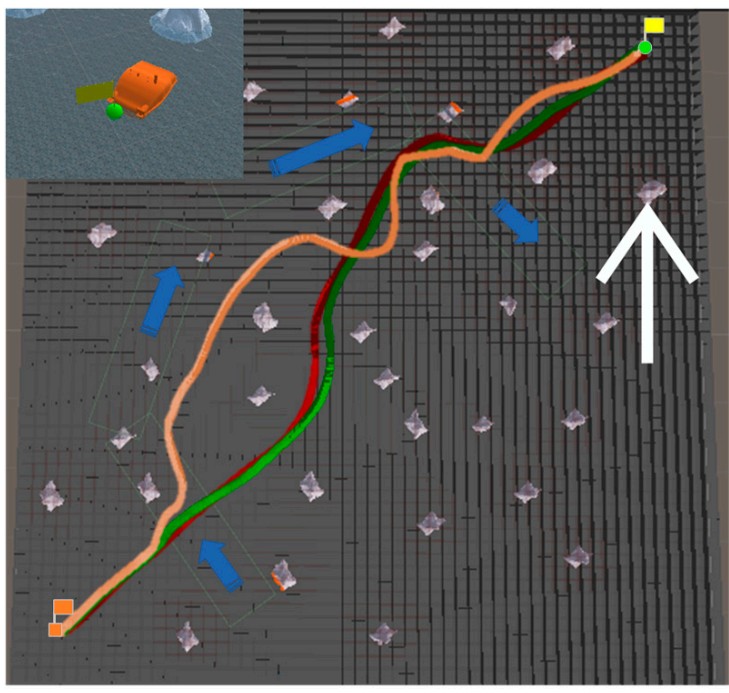

**Figure 15.** Results of the simulation experiment for path tracking.

In the path-tracking simulation experiment, the co-ordinates of the start point were $(-430, -430)$, the co-ordinates of the target point were $(430, 430)$, and the light bule rectangular frame in the map is the wave disturbance area. In order to verify the effectiveness of the proposed algorithm under environmental disturbance, the VPF-RRT* algorithm with a PI was selected as the comparison algorithm, and we compared it with the proposed algorithm to verify the effectiveness of the proposed algorithm under environmental disturbance. The experimental result is shown in Figure 15 and Table 8. The red path is the planning path, and the green track is the actual navigation track of the USV with the proposed algorithm. The orange track is the actual navigation track of the USV with the VPF-RRT* algorithm with a PI controller.

**Table 8.** Results of the simulation experiment for path tracking.

| Algorithm Name | Navigating Time (s) | Navigating Length (m) | Total Turning Angle |
|---|---|---|---|
| The VPF-RRT* algorithm with PI controller | 71.18744 | 1664.6812 | 381°0250′ |
| The proposed algorithm | 62.86429 | 1598.489 | 206°7511′ |

We combined the experimental results in Table 8. As shown by the orange track in Figure 15, after the USV traveled to the disturbance area, due to the influence of waves, the USV deviated from the initial planning path. After replanning the planning path, the USV reselected the planning path that was more inclined to the direction of the $Y_e$-axis, which prolonged the navigating time and increased the navigating length of the USV. As shown in the green track in Figure 14, the proposed algorithm used the anti-environmental disturbance method based on a DRNN-PI controller such that the USV could navigate a planned path in a disturbed area. Therefore, the navigating time of the proposed algorithm is shorter than those of the comparison algorithms, and the navigating length of the proposed algorithm was shorter than that of the comparison algorithm. Therefore, the performance of the proposed algorithm on the map with environmental disturbances was better than that of the comparison algorithms.

In order to avoid the contingency of the experiment results, three groups of different start and end points were selected in a simulation experiment for path tracking. The results are shown in Table 9.

**Table 9.** Results of different start and target points under simulation experiment for path tracking.

| Start and Target Point | Algorithm Name | Navigating Time (s) | Navigating Length (m) | Total Turning Angle |
|---|---|---|---|---|
| (420, −50) (−400, 0) | The VPF-RRT* algorithm with PI controller | 52.15899 | 1040.41123 | 159°3235′ |
| | The proposed algorithm | 43.70968 | 1009.45018 | 120°9144′ |
| (400, 400) (−400, −400) | The VPF-RRT* algorithm with PI controller | 73.15134 | 1632.12661 | 327°4137′ |
| | The proposed algorithm | 59.08492 | 1601.4809 | 291°1008′ |
| (400, 400) (−44, −131) | The VPF-RRT* algorithm with PI controller | 64.15184 | 1303.02153 | 186°5998′ |
| | The proposed algorithm | 41.21003 | 927.60188 | 154°3234′ |

In order to avoid contingency in the experimental results for the total turning angle, 20 groups of different starting points and targets were selected for the simulation experiment, and the results are shown in Figure 16.

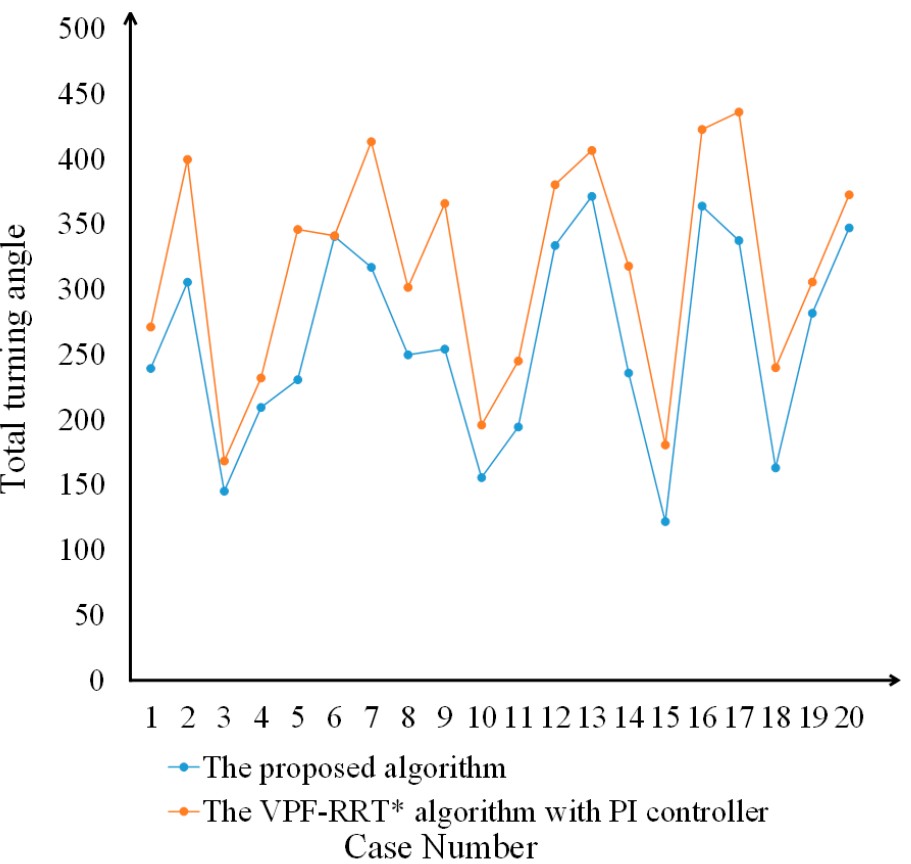

**Figure 16.** Experimental data of 20 groups of different starting points and targets.

It can be seen from Figure 16 that the total turning angle of the proposed algorithm is between 120 and 370, which is lower than the comparison algorithm. Although the total turning angle of the proposed algorithm is similar to that of the comparison algorithm in the sixth experiment, the difference is within a reasonable range. The simulation results show that the proposed algorithm has better control over the USV and has stronger practicality.

## 6. Conclusions

In this paper, a path-planning method that considers environmental disturbances based on VPF-RRT* is proposed. First, on the basis of the RRT* algorithm, a VPF-RRT* algorithm is proposed that can conduct a planning path. Second, an anti-environmental disturbance method based on a DRNN-PI controller is proposed, allowing the USV to eliminate environmental disturbances and still track the planning path. The comparative simulation experiments between the proposed algorithm and other algorithms were conducted within two different experimental scenes. In the path-planning simulation experiment, we compared VPF-RRT* to the RRT* algorithm; the VPF-RRT* algorithm performed a shorter planning path and a smaller total turning angle, which is conducive to USV path tracking. In the path-tracking simulation experiment, the USV using the proposed algorithm could effectively compensate for the impact of environmental disturbances and keep navigating along the planning path. In order to avoid the contingency of the experiment and verify the effectiveness and generality of the proposed algorithm, three experiments were conducted. The experimental results show that the path length of the proposed algorithm was better than other algorithms, and the USV using the proposed algorithm could effectively overcome the impact of environmental disturbances.

In future research, the proposed algorithm might be improved in two aspects. First, most of the parameters in the proposed algorithm were obtained by experience, and these would have been influenced by subjective factors. These parameters can be optimized by other

methods in the future. Second, this paper does not consider the traversal method of multitarget points. In future research, other algorithms will be introduced to determine the traversal order between multitarget points. In addition, the algorithm proposed in this paper has not been tested on a real USV and will be tested on a real USV in the future.

**Author Contributions:** Conceptualization, Z.C. and J.Y.; methodology, Z.C.; software, Z.C.; validation, J.Y., Z.Z. and X.W.; formal analysis, Y.C.; investigation, Z.C.; resources, J.Y.; data curation, J.Y.; writing—original draft preparation, J.Y.; writing—review and editing, X.W.; visualization, X.W.; supervision, Z.Z.; project administration, J.Y.; funding acquisition, Z.Z. All authors have read and agreed to the published version of the manuscript.

**Funding:** This research was funded by the Beijing Talents Project, grant number 2020A28. National Key Research and Development Program of China, grant number 2022YFF1101103.

**Institutional Review Board Statement:** Not applicable.

**Informed Consent Statement:** Not applicable.

**Data Availability Statement:** No new data were created.

**Conflicts of Interest:** The authors declare no conflict of interest.

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
