# Peer review of "A Path-Planning Method Considering Environmental Disturbance Based on VPF-RRT*"

_drones, doi:10.3390/drones7020145_

Round 1
Reviewer 1 Report
The paper considers a path planning method based on a modified RRT algorithm. In the introductory part of the paper, the authors provide a broad overview of the traditional path plannning algorithms, unsuitable in the case of surface vehicles. Tha major problem is smoothing of the path, to make the turning points feasible, and tractable. Apart the obvious reasons for picking up an appropriate method, like path length, feasibility, the calculation time is also a factor. Of course at this point the major question is why have not the authors decide to use Splines?
Contribtions are clearly stated, though a novelty statement could be added as well.
The next section presents the model of the USV, along with a formulation of the problem. Figure 2 - what is a '3D USV'? Add 'model', please.
Section 3.1 - as per random properties of the method, or a random search to be exact, the question is - could you possibly replace the random part by SPSA (Simultaneous Perturbation Stochastic Approximation) of JC Spall? What method is actually used here to provide 'randomness'?
Figure 8 presents an academic form of the PI controller. Have you really not implemented any anti-windup schemes, moved P term to the feedback, or limited the actuator (not mentioning introduction of costraints on the control input)?
Table 2 - please provide some statistics on the perfoemance between RRT and VPF-RRT for a set of randomly picked initial starting points to execute a zero-order hypothesis verification. Some measure of performance is path length divided by the Euclidean distance between A and B points (normalization).
How to PI gains affect the cumulative turning angles? Could you provide a 3D plot based on a grid of selected gains averaged over say 10 simulations each as cumulative turning angle vs/ Kp, Ti?
Author Response
Response to Reviewers of Manuscript drones-2199326
Zhihao Chen, Jiabin Yu*, Zhiyao Zhao, Xiaoyi Wang and Yang Chen
To Reviewer
Reviewer:
- The paper considers a path planning method based on a modified RRT algorithm. In the introductory part of the paper, the authors provide a broad overview of the traditional path plannning algorithms, unsuitable in the case of surface vehicles. Tha major problem is smoothing of the path, to make the turning points feasible, and tractable. Apart the obvious reasons for picking up an appropriate method, like path length, feasibility, the calculation time is also a factor. Of course at this point the major question is why have not the authors decide to use Splines?
Response: Thank you for your valuable advice. In section 5.2, we added a comparative test of the B-spline curve-RRT* with PI controller. The experimental results show that the performance of the B-spline curve-RRT* with PI controller is poor in complex environments.
In simulation experiment of path planning, the coordinate of the start point was (- 423, - 440) and the coordinate of the target point is (411, 411). Based on the above map parameters, the RRT* algorithm with PI controller and the B-spline curve-RRT* with PI controller were selected as the comparison algorithm. The comparison experiment was conducted with the RRT* algorithm with PI controller, the B-spline curve-RRT* with PI controller and the VPF-RRT* algorithm with PI controller to verify the effectiveness of the planning path of VPF-RRT* algorithm. The experimental results were shown in Figure 12, Figure 13, Figure 14 and Table 5.

Figure 12. Results of RRT* algorithm with PI controller in simulation experiment of path planning.

Figure 13. Results of B-spline curve-RRT* with PI controller in simulation experiment of path planning.

Figure 14. Results of VPF-RRT* algorithm with PI controller in simulation experiment of path planning.
Table 5. Results in simulation experiment of path planning.
|
Algorithm name |
Path length(m) |
Total turning angle |
Computational time (s) |
Sailing time (s) |
Normalization index |
|
RRT* with PI controller |
1710.15182 |
161°3752′ |
22.15748 |
182.54843 |
1.43525 |
|
B-spline curve-RRT* with PI controller |
1952.94236 |
286°1475′ |
26.22971 |
201.21855 |
1.63901 |
|
VPF-RRT* with PI controller |
1623.07952 |
89°1398′ |
22.87613 |
178.33216 |
1.36217 |
Combined with the experimental results in Table 5, and the normalized index value is path length divided by the Euclidean distance between start point and target point. It can be seen from Figure 12 that the length of the planning path of the RRT* algorithm with PI controller is longer than the length of the VPF-RRT* algorithm with PI controller. And the total turning angle of the RRT* algorithm with PI controller is also larger than the total turning angle of the VPF-RRT* algorithm with PI controller. It can be seen from Figure 13 that the length of the planning path of the B-spline curve-RRT* with PI controller is longer than the length of the VPF-RRT* algorithm with PI controller. And the total turning angle of the B-spline curve-RRT* with PI controller is also larger than the total turning angle of the VPF-RRT* algorithm with PI controller. As can be seen from Figure 14, since VPF-RRT* algorithm adjusts the position of path nodes based on virtual potential field, which shortens the length of the planning path in VPF-RRT* algorithm and reduces the total turning angle in VPF-RRT* algorithm, and the planning path of VPF-RRT* algorithm was suitable for the motion ability of the USV.
In order to avoid the contingency of the experiment results, three groups of different start points and target points are selected in simulation experiment of path planning for the simulation experiments. The result as shown in Table 6.
Table 6. Results of different start and target points under simulation experiment of path planning.
|
Start and target |
Algorithm name |
Path length(m) |
Total turning angle |
Computational time (s) |
Sailing time (s) |
Normalization index |
|
(-94, -27) (347,-451) |
RRT* algorithm with PI controller |
725.15104 |
45°1268′ |
13.20158 |
90.12177 |
1.11500 |
|
B-spline curve-RRT* with PI controller |
816.20583 |
50°3288′ |
14.98402 |
99.37544 |
1.25501 |
|
|
VPF-RRT* algorithm with PI controller |
709.13587 |
44°1534′ |
13.81035 |
87.79353 |
1.09037 |
|
|
(400, 400) (-400,-400) |
RRT* algorithm with PI controller |
1634.98075 |
109°8418′ |
20.56508 |
171.34998 |
1.44513 |
|
B-spline curve-RRT* with PI controller |
2120.18598 |
302°9211′ |
26.30054 |
219.44503 |
1.87399 |
|
|
VPF-RRT* algorithm with PI controller |
1568.27831 |
105°7518′ |
18.60874 |
168.47056 |
1.38617 |
|
|
(400, -400) (-400, 400) |
RRT* algorithm with PI controller |
1612.83788 |
141°4894′ |
19.32007 |
182.21533 |
1.42556 |
|
B-spline curve-RRT* with PI controller |
1984.33276 |
291°0233′ |
24.11328 |
203.55983 |
1.75391 |
|
|
VPF-RRT* algorithm with PI controller |
1567.15489 |
134°4537′ |
18.83021 |
177.00357 |
1.38518 |
- Contribtions are clearly stated, though a novelty statement could be added as well.
Response: Thank you for your valuable advice. We added novelty statement at the end of the first section. This paper has strong novelty in the field of USV cruise. This paper optimizes the RRT* algorithm to improve the efficiency of path planning. At the same time, this paper improves the PI controller and uses DRNN algorithm to adjust the parameters of the PI controller.
- The next section presents the model of the USV, along with a formulation of the problem. Figure 2 - what is a '3D USV'? Add 'model', please.
Response: Thank you for your valuable advice. We have modified the "3D USV" in Figure 3 and have corrected it to "3D USV model"
- Section 3.1 - as per random properties of the method, or a random search to be exact, the question is - could you possibly replace the random part by SPSA (Simultaneous Perturbation Stochastic Approximation) of JC Spall? What method is actually used here to provide 'randomness'?
Response: Thank you for your valuable advice. In section 4.1, we added the description of RRT algorithm sampling random points. RRT algorithm performs random sampling in a limited sampling space, the size of which is determined by the sampling range. Divide the sampling space into n parts, and then randomly select a point as a random sampling point. The probability of selecting this point is 1/n. RRT algorithm is an incremental construction method. During the construction process, the algorithm continuously generates a random state point in the search space. If the point is in a collision-free position, the nearest node in the search tree is found as the reference node. From the reference node, it extends towards the random node with a certain step length. The location of the end point of the extension line is added to the search tree as an effective node. The growth process of the search tree continues until the distance between the target node and the search tree is within a certain range. Then the search algorithm searches for the shortest path connecting the start point to the end point in the search tree.
- Figure 9 presents an academic form of the PI controller. Have you really not implemented any anti-windup schemes, moved P term to the feedback, or limited the actuator (not mentioning introduction of costraints on the control input)?
Response: Thank you for your valuable advice. We have modified the content in Figure 9 and added a description at the bottom of Figure 9, using the integral separation method and setting the dead zone to suppress the integral saturation. In Figure 9, the expected included angle between the forward direction of the USV and the forward direction of planning path is equal to the compensation angle, i. e. Ψd(t) = α(t). The error e(t) is the difference between the expected included angle Ψd(t) and the actual included angle Ψ(t), i. e. e(t) = α(t) - Ψ(t). The integral separation is used to solve the integral saturation problem of PI controller. When the integral is about to reach the saturation state, set the dead zone to limit the integral to reach the saturation state. When integral saturation occurs, integral separation is used to eliminate integral saturation.
- Table 2 - please provide some statistics on the perfoemance between RRT and VPF-RRT for a set of randomly picked initial starting points to execute a zero-order hypothesis verification. Some measure of performance is path length divided by the Euclidean distance between A and B points (normalization).
Response: Thank you for your valuable advice. We set a normalized index, whose value is path length divided by the Euclidean distance between start point and target point.
Table 5. Results in simulation experiment of path planning.
|
Algorithm name |
Path length(m) |
Total turning angle |
Computational time (s) |
Sailing time (s) |
Normalization index |
|
RRT* with PI controller |
1710.15182 |
161°3752′ |
22.15748 |
182.54843 |
1.43525 |
|
B-spline curve-RRT* with PI controller |
1952.94236 |
286°1475′ |
26.22971 |
201.21855 |
1.63901 |
|
VPF-RRT* with PI controller |
1623.07952 |
89°1398′ |
22.87613 |
178.33216 |
1.36217 |
Combined with the experimental results in Table 5, and the normalized index value is path length divided by the Euclidean distance between start point and target point. It can be seen from Figure 12 that the length of the planning path of the RRT* algorithm with PI controller is longer than the length of the VPF-RRT* algorithm with PI controller. And the total turning angle of the RRT* algorithm with PI controller is also larger than the total turning angle of the VPF-RRT* algorithm with PI controller. It can be seen from Figure 13 that the length of the planning path of the B-spline curve-RRT* with PI controller is longer than the length of the VPF-RRT* algorithm with PI controller. And the total turning angle of the B-spline curve-RRT* with PI controller is also larger than the total turning angle of the VPF-RRT* algorithm with PI controller. As can be seen from Figure 14, since VPF-RRT* algorithm adjusts the position of path nodes based on virtual potential field, which shortens the length of the planning path in VPF-RRT* algorithm and reduces the total turning angle in VPF-RRT* algorithm, and the planning path of VPF-RRT* algorithm was suitable for the motion ability of the USV.
In order to avoid the contingency of the experiment results, three groups of different start points and target points are selected in simulation experiment of path planning for the simulation experiments. The result as shown in Table 6.
Table 6. Results of different start and target points under simulation experiment of path planning.
|
Start and target |
Algorithm name |
Path length(m) |
Total turning angle |
Computational time (s) |
Sailing time (s) |
Normalization index |
|
(-94, -27) (347,-451) |
RRT* algorithm with PI controller |
725.15104 |
45°1268′ |
13.20158 |
90.12177 |
1.11500 |
|
B-spline curve-RRT* with PI controller |
816.20583 |
50°3288′ |
14.98402 |
99.37544 |
1.25501 |
|
|
VPF-RRT* algorithm with PI controller |
709.13587 |
44°1534′ |
13.81035 |
87.79353 |
1.09037 |
|
|
(400, 400) (-400,-400) |
RRT* algorithm with PI controller |
1634.98075 |
109°8418′ |
20.56508 |
171.34998 |
1.44513 |
|
B-spline curve-RRT* with PI controller |
2120.18598 |
302°9211′ |
26.30054 |
219.44503 |
1.87399 |
|
|
VPF-RRT* algorithm with PI controller |
1568.27831 |
105°7518′ |
18.60874 |
168.47056 |
1.38617 |
|
|
(400, -400) (-400, 400) |
RRT* algorithm with PI controller |
1612.83788 |
141°4894′ |
19.32007 |
182.21533 |
1.42556 |
|
B-spline curve-RRT* with PI controller |
1984.33276 |
291°0233′ |
24.11328 |
203.55983 |
1.75391 |
|
|
VPF-RRT* algorithm with PI controller |
1567.15489 |
134°4537′ |
18.83021 |
177.00357 |
1.38518 |
- How to PI gains affect the cumulative turning angles? Could you provide a 3D plot based on a grid of selected gains averaged over say 10 simulations each as cumulative turning angle vs/ Kp, Ti?
Response: Thank you for your valuable advice. In order to avoid the contingency of the experiment results, three groups of different start points and end points are selected in simulation experiment of path tracking for simulation experiments. The result as shown in Table 9. In order to avoid contingency in the experimental results of the total turning angle, 20 groups of different starting points and targets were selected for the simulation experiment, and the results were shown in Fig. 16.

Figure 16. Experimental data of 20 groups of different starting points and targets.
It can be seen from Figure 16 that the total turning angle of the proposed algorithm is between 120 and 370, which is lower than the comparison algorithm. Although the total turning angle of the proposed algorithm is similar to that of the comparison algorithm in the sixth experiment, the difference is within a reasonable range. The simulation results show that the proposed algorithm has better control effect on USV and has strong practicability.

Reviewer 2 Report
In the traditional fast exploratory random tree (RRT) algorithm, the planned path is not smooth, the distance is large and the fault tolerance of the planned path is low. At the same time, there are perturbations in the environment that cause the unmanned surface vessel (USV) to deviate from the planned path during navigation. Therefore, the authors have proposed a path planning method to account for environmental perturbations based on virtual potential field RRT* (VPF-RRT*). Firstly, based on the RRT* algorithm, a VPF-RRT* algorithm for path planning is proposed. Secondly, a method for dealing with environmental perturbations based on deep recurrent neural network PI (DRNN-PI) controller is proposed so that USV can remove environmental perturbations and track the planning path. Comparative simulation experiments between the proposed algorithm and other algorithms are conducted in two different experimental scenes. In the path planning simulation experiment, compared to the RRT * algorithm, the VPF-RRT * algorithm has a shorter planning path and a smaller overall turning angle.
The novelty of this work is questionable to me.
Using a neural network as a PID tuner is a fairly well-known task. In general, PID tuning is a very well known task that is performed by a huge number of methods. The specific link between such tuning and the RRT algorithm is not fully disclosed.
The use of the neural network itself requires a more detailed description of how this network was trained, etc.
The simulation results are too brief and do not look convincing. The description of the models needs disclosure.
Author Response
Response to Reviewers of Manuscript drones-2199326
Zhihao Chen, Jiabin Yu*, Zhiyao Zhao, Xiaoyi Wang and Yang Chen
To Reviewer
Reviewer:
- Using a neural network as a PID tuner is a fairly well-known task. In general, PID tuning is a very well known task that is performed by a huge number of methods. The specific link between such tuning and the RRT algorithm is not fully disclosed.
Response: Thank you for your valuable advice. In the first paragraph of section 4.2, we added the specific link between DRNN-PI controller and RRT algorithm. The section 4.1 introduced the VPF-RRT* algorithm, which is used to plan the path from the starting point to the target point. The section 4.2 proposes an anti-environmental disturbance method based on DRNN-PI controller, which is used to control USV navigation on the planned path and realize path tracking. At the same time, we added an experiment in section 5.1, using BP-PI, CMAC-PI and DRNN-PI controllers for comparative tests. The USV navigates on the path planned by the RRT algorithm, and the DRNN-PI controller is used to control the heading angle of the USV to keep the USV sailing on the planned path.
The use of the neural network itself requires a more detailed description of how this network was trained, etc.
Response: Thank you for your valuable advice. We added a description in section 4.2 to explain how the neural network is trained. We add RMSE and MRE loss functions to monitor the training process of neural network.
The deep recurrent neural network (DRNN) network is used to adjust the performance of PI controller. The hidden layer of neural network has L nodes. There are N samples in each USV heading angle data set. The output of a single hidden layer is as follows:
where si is the output of a single hidden layer node. The DRNN neural network aims to minimize the error loss between the expected heading angle and the actual heading angle as shown in follow:
The loss function is used to obtain the output error of the network model, and the gradient descent method is used to find the minimum value, update the weight, and finally make the model converge. Mean Square Error (MSE) is selected as the loss function to measure the deviation between the predicted value and the real value after each step of training in the neural network. During the operation, the output value of the loss function will become smaller and smaller, and eventually approach zero. The expression of MSE is as follows:
where Ψd (t) is the expected heading angle, Ψ (t) is the actual heading angle, and N is the total number of samples. In addition, the optimizer selects the Root Mean Square Propagation (RMSProp) method. The RMSProp method is suitable for dealing with non-stationary targets. Root Mean Square Error (RMSE) and Mean Relative Error (MRE) are used to verify the prediction accuracy of the neural network model. The expression is as follows:
where Ψd (t) is the expected heading angle, Ψ (t) is the actual heading angle, and N is the total number of samples.
The simulation results are too brief and do not look convincing. The description of the models needs disclosure.
Response: Thank you for your valuable advice. In the experiment in section 5.1, we verified the effectiveness of DRNN-PI controller in controlling the heading angle of USV.
Set a start point and a target point in an area containing obstacles, and use RRT algorithm to plan the path from the start point to the target point. The USV advances at a speed of 10m/s, and uses DRNN-PI controller to control the USV's steering. The DRNN neural network is trained based on the principle of the minimum difference between the actual heading angle and the expected heading angle of the USV. The simulation experiment is conducted in the obstacle environment. BP and CMAC algorithm are used as comparison algorithms, and the error curves of the neural network are shown in Figure 10.

Figure 10. Simulation experiment results in obstacle environment.
It can be seen from Figure 10 that the convergence speed of BP and CMAC algorithms are slower than that of the DRNN algorithm. The error curve of BP algorithm tends to be stable only when the step of iterations reaches about 1300. The error curve of the CMAC algorithm tends to be stable only when the step of iterations reaches about 1200. When the DRNN algorithm iterates to about 1000 steps, the error curve starts to stabilize. Therefore, the convergence speed of the DRNN algorithm in the obstacle environment is higher than that of the comparison algorithm, and the loss value of the DRNN algorithm is lower.
BP, CMAC and DRNN network models were established. The network model parameters are set and trained for 2000 times respectively. The loss function value and computational time after the program running were recorded. The comparison results of control accuracy of different models were shown in Table 4.
Table 4. Control accuracy of different models.
|
Algorithm |
RMSE |
MRE |
Loss |
Computational time (s) |
|
BP-PI |
7.3519 |
12%-14% |
0.0396 |
18.5181 |
|
CMAC-PI |
6.5987 |
8%-11% |
0.0287 |
16.2566 |
|
DRNN-PI |
5.9182 |
7%-9% |
0.0211 |
16.3271 |
Table 4 shows that compared with BP and CMAC, RMSE, MRE and Loss of DRNN networks are lower. Because the hidden layer structure of the DRNN is more complex, the computational time of the DRNN is long, but within a reasonable range, and the control accuracy of the DRNN is higher, so the effectiveness of the DRNN is better. The computational time refers to the difference between the sailing time obtained after the planning path length is divided by 10m/s and the actual sailing time of the USV, that is, the time consumed by the PI controller to adjust the heading angle of the USV.
In order to avoid contingency in the experimental results, 20 groups of different starting points and targets were selected for the simulation experiment, and the results were shown in Figure 11.

Figure 11. Experimental data of 20 groups of different starting points and targets.
It can be seen from Figure 11 that during the whole USV sailing process, the computational time required for the DRNN-PI algorithm to adjust USV heading angle was between 15s and 16s, which was lower than the comparison algorithm. Although in the 14th and 19th experiments, the steps required by the proposed algorithm were similar with those of the comparison algorithm, the difference was within a reasonable range. Therefore, the simulation experiment could prove that the ability to control USV of the DRNN-PI algorithm was high and had strong practicability.

Reviewer 3 Report
Summary: In this paper, a virtual potential field RRT* (VPF-RRT*)-based path planning method that takes environmental disturbance into account is proposed. A VPF-RRT* method is first presented to plan the planning path based on the RRT* algorithm. Second, a deep recurrent neural network PI (DRNN-PI) controller-based anti-environmental disturbance technique is suggested to enable the USV to eliminate environmental disturbance and follow the planned course. Two distinct experimental situations are used for the comparison simulation studies between the proposed algorithm and existing algorithms. Comments and Suggestions: - Please, use the capitalized form for the title. - It is preferable to avoid the use of abbreviations in the title. - Pay attention to English mistakes: e.g., Abstract: proposed to *makes* the USV - Thorough proofreading is required for the entire paper. - It is preferable to split the introduction into two parts: Introduction + Related Works - A new figure illustrating the proposed approach can be inserted between sections 1 and 2. - The related works section may be summarized in tabular form. - The authors are invited to include the following interesting references (and others) in their study: 1. https://link.springer.com/article/10.1007/s10619-019-07276-9 2. https://www.sciencedirect.com/science/article/abs/pii/S016781910000048X 3. https://ieeexplore.ieee.org/stamp/stamp.jsp?arnumber=9938438- In addition, the authors are invited to report on the use of well-known formal methods for checking and guaranteeing the correctness of IoT-based systems.
- For this purpose, they may consider the following references (and others):
1. https://link.springer.com/article/10.1007/s11036-019-01369-6
2. https://link.springer.com/chapter/10.1007/978-3-642-24270-0_17
3. https://www.sciencedirect.com/science/article/pii/S1389128619317116 - Lines 190-191: "Therefore, it is necessary to adjust the path node and use the VPFRRT * algorithm to plan the path of USV" ===> Please offer more justifications for selecting this algorithm. - Figure 4: The title and the figure do not appear on the same page. - Please provide more information and an explanation for equations 9 and 11. - Algorithm 1: Is it possible to estimate the complexity of this algorithm and provide proof of its correctness? - Figure 5: Does the considered network have only one hidden layer? - Line 226: Please argue more about the choice of DRNN? Is it possible to consider other types of networks?- Which dataset was used for training the considered neural network? Did you use a pre-trained model? - Is it possible to share the code used in the experiments? - Why not achieve real experiments using real USVs? The authors must identify the limitations of their approach and propose new work directions for the future.
Author Response
Response to Reviewers of Manuscript drones-2199326
Zhihao Chen, Jiabin Yu*, Zhiyao Zhao, Xiaoyi Wang and Yang Chen
To Reviewer
Reviewer:
- Please, use the capitalized form for the title.
Response: Thank you for your valuable advice. We corrected the title and use the capitalized form for the title.
- It is preferable to avoid the use of abbreviations in the title.
Response: Thank you for your valuable advice. We will avoid using abbreviations in the title in future research.
- Pay attention to English mistakes: e.g., Abstract: proposed to *makes* the USV. Thorough proofreading is required for the entire paper.
Response: Thank you for your valuable advice. We have corrected the corresponding contents.
- It is preferable to split the introduction into two parts: Introduction + Related Works.
Response: Thank you for your valuable advice. We had split the introduction into two parts: Introduction + Related Works.
- A new figure illustrating the proposed approach can be inserted between sections 1 and 2.
Response: Thank you for your valuable advice. We have inserted the new figure illustrating the proposed algorithm between sections 1 and 2. The implementation process of the algorithm is shown in Figure 1.

Figure 1. The path planning method considering environmental disturbance based on VPF-RRT* flowchart.
- The related works section may be summarized in tabular form.
Response: Thank you for your valuable advice. We have summarized the related works in tabular form.
Table 1. The algorithm and simulation experimental parameters.
|
Algorithm |
Path smoothing |
Increase of efficiency |
|
Autonomous land vehicle path planning algorithm based on improved heuristic function of A-Star [6] |
Yes |
/ |
|
Improved Safety-First A-Star Algorithm for Autonomous Vehicle [7] |
Yes |
/ |
|
Multi-agent trajectory planning: A decentralized iterative algorithm based on single-agent dynamic RRT star [8] |
/ |
Yes |
|
Boundary-RRT* Algorithm for Drone Collision Avoidance and Interleaved Path Re-planning [9] |
/ |
Yes |
Table 2. The algorithm and simulation experimental parameters.
|
Algorithm |
Improve anti-interference capability |
Increase flexibility |
|
Fractional-Order Controller for Course-Keeping of Underactuated Surface Vessels Based on Frequency Domain Specification and Improved Particle Swarm Optimization Algorithm [10] |
Yes |
/ |
|
Anti-disturbance leader–follower synchronization control of marine vessels for underway replenishment based on robust exact differentiators [11] |
Yes |
/ |
|
An anti-vibration-shock inertial matching measurement method for hull deformation [12] |
Yes |
/ |
|
Anti-disturbance control for dynamic positioning system of ships with disturbances [13] |
Yes |
/ |
|
Output-Feedback Flocking Control of Multiple Autonomous Surface Vehicles Based on Data-Driven Adaptive Extended State Observers [14] |
Yes |
/ |
|
Research and Comparison of Automatic Control Algorithm for Unmanned Ship [15] |
/ |
Yes |
|
Backstepping-Based Controller Design for Uncertain Switched High-Order Nonlinear Systems via PI Compensation [16] |
/ |
Yes |
- The authors are invited to include the following interesting references (and others) in their study: 1. https://link.springer.com/article/10.1007/s10619-019-07276-9 2. https://www.sciencedirect.com/science/article/abs/pii/S016781910000048X 3. https://ieeexplore.ieee.org/stamp/stamp.jsp?arnumber=9938438
- In addition, the authors are invited to report on the use of well-known formal methods for checking and guaranteeing the correctness of IoT-based systems.
- For this purpose, they may consider the following references (and others):
- https://link.springer.com/article/10.1007/s11036-019-01369-6
- https://link.springer.com/chapter/10.1007/978-3-642-24270-0_17
- https://www.sciencedirect.com/science/article/pii/S1389128619317116
Response: Thank you for your valuable advice. We have referred to the references you provided and revised and optimized our article.
- Lines 190-191: "Therefore, it is necessary to adjust the path node and use the VPFRRT * algorithm to plan the path of USV" ===> Please offer more justifications for selecting this algorithm.
Response: Thank you for your valuable advice. We added more justifications for selecting this algorithm. When the distance between the planning path and the obstacle is close, the fault tolerance rate of the planning path will be reduced, which is not conducive to the subsequent path tracking of USV. The smooth planning path is more suitable for the path tracking of USV, but there are many unnecessary inflection points in the planning path of the traditional RRT * algorithm, so it is necessary to adjust the nodes to optimize the angle of in-flection points. However, the existing path smoothing algorithm will increase the memory loss of the algorithm. Therefore, it is necessary to adjust the path node and use the VPF-RRT * algorithm to plan the path of USV. The expansion step is p, the initial value of i is 1, the maximum number of iterations is n.
- Figure 4: The title and the figure do not appear on the same page.
Response: Thank you for your valuable advice. We adjusted the layout of the article.
- Please provide more information and an explanation for equations 9 and 11.
Response: Thank you for your valuable advice. We have corrected the corresponding contents.
Let the repulsive force of the obstacle on the node be Fc, and its expression is as follows:
,
where η is the repulsion coefficient, whose value is set manually; ωmax is the maximum angular velocity of USV; Rmin is the minimum turning radius of USV; ρ is the range of influence of obstacles on nodes. When the path node exceeds the influence range of the obstacle, the repulsive force is zero.
Use Fy* to calculate the compensation distance ΔL. The compensation distance ΔL will be used to adjust the position of the path node, and its expression is as follows:
.
- Algorithm 1: Is it possible to estimate the complexity of this algorithm and provide proof of its correctness?
Response: Thank you for your valuable advice. We added experiments to provide proof of the proposed algorithm correctness, and the complexity of the proposed algorithm is compared with that of other algorithms.
In simulation experiment of path planning, the coordinate of the start point was (- 423, - 440) and the coordinate of the target point is (411, 411). Based on the above map parameters, the RRT* algorithm with PI controller and the B-spline curve-RRT* with PI controller were selected as the comparison algorithm. The comparison experiment was conducted with the RRT* algorithm with PI controller, the B-spline curve-RRT* with PI controller and the VPF-RRT* algorithm with PI controller to verify the effectiveness of the planning path of VPF-RRT* algorithm. The experimental results were shown in Figure 12, Figure 13, Figure 14 and Table 3.

Figure 12. Results of RRT* algorithm with PI controller in simulation experiment of path planning.

Figure 13. Results of B-spline curve-RRT* with PI controller in simulation experiment of path planning.

Figure 14. Results of VPF-RRT* algorithm with PI controller in simulation experiment of path planning.
Table 3. Results in simulation experiment of path planning.
|
Algorithm name |
Path length(m) |
Total turning angle |
Computational time (s) |
Sailing time (s) |
Normalization index |
|
RRT* with PI controller |
1710.15182 |
161°3752′ |
22.15748 |
182.54843 |
1.43525 |
|
B-spline curve-RRT* with PI controller |
1952.94236 |
286°1475′ |
26.22971 |
201.21855 |
1.63901 |
|
VPF-RRT* with PI controller |
1623.07952 |
89°1398′ |
22.87613 |
178.33216 |
1.36217 |
Combined with the experimental results in Table 3, and the normalized index value is path length divided by the Euclidean distance between start point and target point. It can be seen from Figure 12 that the length of the planning path of the RRT* algorithm with PI controller is longer than the length of the VPF-RRT* algorithm with PI controller. And the total turning angle of the RRT* algorithm with PI controller is also larger than the total turning angle of the VPF-RRT* algorithm with PI controller. It can be seen from Figure 13 that the length of the planning path of the B-spline curve-RRT* with PI controller is longer than the length of the VPF-RRT* algorithm with PI controller. And the total turning angle of the B-spline curve-RRT* with PI controller is also larger than the total turning angle of the VPF-RRT* algorithm with PI controller. As can be seen from Figure 14, since VPF-RRT* algorithm adjusts the position of path nodes based on virtual potential field, which shortens the length of the planning path in VPF-RRT* algorithm and reduces the total turning angle in VPF-RRT* algorithm, and the planning path of VPF-RRT* algorithm was suitable for the motion ability of the USV.
In order to avoid the contingency of the experiment results, three groups of different start points and target points are selected in simulation experiment of path planning for the simulation experiments. The result as shown in Table 4.
Table 4. Results of different start and target points under simulation experiment of path planning.
|
Start and target |
Algorithm name |
Path length(m) |
Total turning angle |
Computational time (s) |
Sailing time (s) |
Normalization index |
|
(-94, -27) (347, -451) |
RRT* algorithm with PI controller |
725.15104 |
45°1268′ |
13.20158 |
90.12177 |
1.11500 |
|
B-spline curve-RRT* with PI controller |
816.20583 |
50°3288′ |
14.98402 |
99.37544 |
1.25501 |
|
|
VPF-RRT* algorithm with PI controller |
709.13587 |
44°1534′ |
13.81035 |
87.79353 |
1.09037 |
|
|
(400, 400) (-400,-400) |
RRT* algorithm with PI controller |
1634.98075 |
109°8418′ |
20.56508 |
171.34998 |
1.44513 |
|
B-spline curve-RRT* with PI controller |
2120.18598 |
302°9211′ |
26.30054 |
219.44503 |
1.87399 |
|
|
VPF-RRT* algorithm with PI controller |
1568.27831 |
105°7518′ |
18.60874 |
168.47056 |
1.38617 |
|
|
(400, -400) (-400, 400) |
RRT* algorithm with PI controller |
1612.83788 |
141°4894′ |
19.32007 |
182.21533 |
1.42556 |
|
B-spline curve-RRT* with PI controller |
1984.33276 |
291°0233′ |
24.11328 |
203.55983 |
1.75391 |
|
|
VPF-RRT* algorithm with PI controller |
1567.15489 |
134°4537′ |
18.83021 |
177.00357 |
1.38518 |
- Figure 5: Does the considered network have only one hidden layer?
Response: Thank you for your valuable advice. We have corrected the corresponding contents. The DRNN network includes one input layer, two hidden layers and one output layer.

Figure 6. Structure of DRNN.
- Line 226: Please argue more about the choice of DRNN? Is it possible to consider other types of networks?
Response: Thank you for your valuable advice. We have corrected the corresponding contents, and a experiments were added in Section 5.1. The recurrent neural networks can be divided into single hidden layer recurrent neural networks and deep recurrent neural networks (DRNN) according to the structure [19]. DRNN can enhance the ability of PI controller, which can repeat the loop body at each time for many times. The parameters in the loop body of each layer of DRNN network are shared, but the parameters between different layers can be different. DRNN network has strong sensitivity. DRNN stores the output value of the hidden layer neuron at the previous time through the receiving layer, and returns it to the input of the first hidden layer, so that the final output of the network is related the current information and the historical information. DRNN has strong dynamic information processing ability, high prediction accuracy, and strong generalization ability, which can avoid local minimum problems.
Set a start point and a target point in an area containing obstacles, and use RRT algorithm to plan the path from the start point to the target point. The USV advances at a speed of 10m/s, and uses DRNN-PI controller to control the USV's steering. The DRNN neural network is trained based on the principle of the minimum difference between the actual heading angle and the expected heading angle of the USV. The simulation experiment is conducted in the obstacle environment. BP and CMAC algorithm are used as comparison algorithms, and the error curves of the neural network are shown in Figure 10.

Figure 10. Simulation experiment results in obstacle environment.
It can be seen from Figure 10 that the convergence speed of BP and CMAC algorithms are slower than that of the DRNN algorithm. The error curve of BP algorithm tends to be stable only when the step of iterations reaches about 1300. The error curve of the CMAC algorithm tends to be stable only when the step of iterations reaches about 1200. When the DRNN algorithm iterates to about 1000 steps, the error curve starts to stabilize. Therefore, the convergence speed of the DRNN algorithm in the obstacle environment is higher than that of the comparison algorithm, and the loss value of the DRNN algorithm is lower.
BP, CMAC and DRNN network models were established. The network model parameters are set and trained for 2000 times respectively. The loss function value and computational time after the program running were recorded. The comparison results of control accuracy of different models were shown in Table 4.
Table 4. Control accuracy of different models.
|
Algorithm |
RMSE |
MRE |
Loss |
Computational time (s) |
|
BP-PI |
7.3519 |
12%-14% |
0.0396 |
18.5181 |
|
CMAC-PI |
6.5987 |
8%-11% |
0.0287 |
16.2566 |
|
DRNN-PI |
5.9182 |
7%-9% |
0.0211 |
16.3271 |
Table 4 shows that compared with BP and CMAC, RMSE, MRE and Loss of DRNN networks are lower. Because the hidden layer structure of the DRNN is more complex, the computational time of the DRNN is long, but within a reasonable range, and the control accuracy of the DRNN is higher, so the effectiveness of the DRNN is better. The computational time refers to the difference between the sailing time obtained after the planning path length is divided by 10m/s and the actual sailing time of the USV, that is, the time consumed by the PI controller to adjust the heading angle of the USV.
In order to avoid contingency in the experimental results, 20 groups of different starting points and targets were selected for the simulation experiment, and the results were shown in Figure 11.

Figure 11. Experimental data of 20 groups of different starting points and targets.
It can be seen from Figure 10 that during the whole USV sailing process, the computational time required for the DRNN-PI algorithm to adjust USV heading angle was between 15s and 16s, which was lower than the comparison algorithm. Although in the 14th and 19th experiments, the steps required by the proposed algorithm were similar with those of the comparison algorithm, the difference was within a reasonable range. Therefore, the simulation experiment could prove that the ability to control USV of the DRNN-PI algorithm was high and had strong practicability.
- Which dataset was used for training the considered neural network? Did you use a pre-trained model?
Response: Thank you for your valuable advice. We added experiments in section 5.1 to illustrate the training process of neural network.
- Is it possible to share the code used in the experiments?
Response: Thank you for your valuable advice. We will improve our code in subsequent research and apply for patents. Therefore, at the current stage, the code needs to be confidential. The code will be published after improvement. Thank you for your understanding.
- Why not achieve real experiments using real USVs? The authors must identify the limitations of their approach and propose new work directions for the future.
Response: Thank you for your valuable advice. We have added corresponding contents to the future plan. The efficiency of the real USV simulation experiment is low, and it can not effectively eliminate the contingency in the simulation experiment. However, we will try to use the real USV for simulation experiments in the follow-up research. In the future research, the proposed algorithm can be improved in two aspects. First, most of the parameters in the proposed algorithm are obtained by experience, and they will be influenced by subjective factors. These parameters can be optimized by some methods in the future. Second, this paper does not consider the traversal method of multi-target points. In future research, other algorithms will be introduced to determine the traversal order between multi-target points. In addition, the algorithm proposed in this paper has not been tested on a real USV, and will be tested on a real USV in the future.

Round 2
Reviewer 1 Report
Thank you for implementing the changes requested. I fully agree with your point of view. The paper is ready for publication now. Good luck with the review process.
Author Response
Response to Reviewers of Manuscript drones-2199326
Zhihao Chen, Jiabin Yu*, Zhiyao Zhao, Xiaoyi Wang and Yang Chen
To Reviewer
We highly appreciate your carefulness, conscientious and the broad knowledge on the relevant research fields, since you have given us a number of beneficial suggestions.
Reviewer 2 Report
The authors have done a lot of revising. Nevertheless, I still have my doubts about the novelty of the work
Author Response
Response to Reviewers of Manuscript drones-2199326
Zhihao Chen, Jiabin Yu*, Zhiyao Zhao, Xiaoyi Wang and Yang Chen
To Reviewer
We highly appreciate your carefulness, conscientious and the broad knowledge on the relevant research fields, since you have given us a number of beneficial suggestions. Traditionally, sampling-based motion planning method performs sampling to cover the state space, and in particular, low-cost path search algorithms such as RRT∗ require a large number of samples to find near-optimal solutions. Due to the complex state space and collision constraints, the computational burden caused by the sampling process can increase. In order to accelerate the planning process of the sample-based motion planning algorithm, we improved the RRT * algorithm by establishing a virtual potential field. The main contribution of this paper is not only to find low-cost trajectories that satisfy specifications, but also to increase the efficiency of the searching process. This paper introduces a virtual potential field to enhance the correlation between the planning path and the environment, which is novel in the field of path planning.

Reviewer 3 Report
The authors took my remarks and suggestions into account.
There is a minor issue with the additional references: the family names of the authors have been abbreviated rather than their first names, as is customary for references.
Author Response
Response to Reviewers of Manuscript drones-2199326
Zhihao Chen, Jiabin Yu*, Zhiyao Zhao, Xiaoyi Wang and Yang Chen
To Reviewer
We highly appreciate your carefulness, conscientious and the broad knowledge on the relevant research fields, since you have given us a number of beneficial suggestions. We have revised the corresponding references. Thank you again.